# STEVE-1: A Generative Model for Text-to-Behavior in Minecraft

**Shalev Lifshitz**[1,2*]
shalev.lifshitz@mail.utoronto.ca

**Keiran Paster**[1,2*]
keirp@cs.toronto.edu

**Harris Chan**[1,2†]
hchan@cs.toronto.edu

**Jimmy Ba**[1,2]
jba@cs.toronto.edu

**Sheila McIlraith**[1,2]
sheila@cs.toronto.edu

[1]Department of Computer Science, University of Toronto, Toronto, Canada.
[2]Vector Institute for Artificial Intelligence, Toronto, Canada.

## Abstract

Constructing AI models that respond to text instructions is challenging, especially for sequential decision-making tasks. This work introduces a methodology, inspired by unCLIP, for instruction-tuning generative models of behavior without relying on a large dataset of instruction-labeled trajectories. Using this methodology, we create an instruction-tuned Video Pretraining (VPT) model called STEVE-1, which can follow short-horizon open-ended text and visual instructions in Minecraft™. STEVE-1 is trained in two steps: adapting the pretrained VPT model to follow commands in MineCLIP's latent space, then training a prior to predict latent codes from text. This allows us to finetune VPT through self-supervised behavioral cloning and hindsight relabeling, reducing the need for costly human text annotations, and all for only $60 of compute. By leveraging pretrained models like VPT and MineCLIP and employing best practices from text-conditioned image generation, STEVE-1 sets a new bar for open-ended instruction following in Minecraft with low-level controls (mouse and keyboard) and raw pixel inputs, far outperforming previous baselines and robustly completing 12 of 13 tasks in our early-game evaluation suite. We provide experimental evidence highlighting key factors for downstream performance, including pretraining, classifier-free guidance, and data scaling. All resources, including our model weights, training scripts, and evaluation tools are made available for further research.

## 1 Introduction

The ability to use text instructions to control and interact with powerful AI models has made these models accessible and customizable for the masses. Such models include ChatGPT [41], which can respond to messages written in natural language and perform a wide array of tasks, and Stable Diffusion [50], which turns natural language into an image. While those models cost anywhere from hundreds of thousands to hundreds of millions of dollars to train, there has been an equally exciting trend whereby powerful open-source foundation models like LLaMA [59] can be finetuned with surprisingly little compute and data to become instruction-following (e.g., [58, 13]).

In this paper, we study whether such an approach could be applicable to sequential decision-making domains. Unlike in text and image domains, diverse data for sequential decision-making is very expensive and often does not come with a convenient "instruction" label like captions for images. We propose to instruction-tune pretrained generative models of behavior, mirroring the advancements seen in recent instruction-tuned LLMs like Alpaca [58], without relying on a large dataset of instruction-labeled trajectories.

---

[*]Equal contribution.
[†]Core contribution.

37th Conference on Neural Information Processing Systems (NeurIPS 2023).

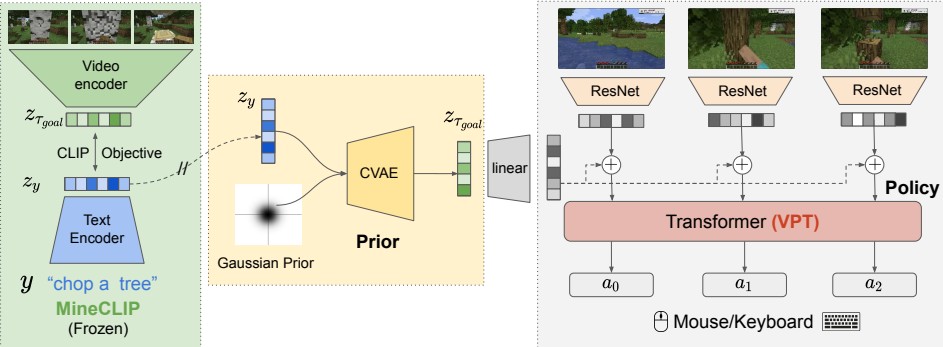

Figure 1: Like unCLIP [48], our approach involves two models. First, we train the *policy* by finetuning VPT to achieve goals given by pretrained MineCLIP [17] visual embeddings using our gameplay dataset. Second, for the *prior* model, we train a CVAE [54] to sample MineCLIP visual embeddings given a text prompt. The combination of these two models enables our agent to follow text and visual instructions.

In the past year, two foundation models for the popular open-ended video game Minecraft™ were released: a foundation model for behavior called VPT [5] and a model aligning text and video clips called MineCLIP [17]. This has opened up an intriguing avenue to explore fine-tuning for instruction-following in the sequential decision-making domain of Minecraft. VPT was trained on 70k hours of Minecraft gameplay, so the agent already has vast knowledge about the Minecraft environment. However, just as the massive potential of LLMs was unlocked by aligning them to follow instructions, it is likely that the VPT model has the potential for general, controllable behavior if it is finetuned to follow instructions. In particular, our paper demonstrates a method for fine-tuning VPT to follow short-horizon text instructions with only $60 of compute and around 2,000 instruction-labeled trajectory segments.

Our method draws inspiration from unCLIP [48], the approach used to create the popular text-to-image model DALL•E 2. We decompose the problem of creating an instruction-following Minecraft agent into two models: a VPT model finetuned to achieve visual goals embedded in the MineCLIP latent space, and a prior model that translates text instructions into MineCLIP visual embeddings. We finetune VPT using behavioral cloning with self-supervised data generated with hindsight relabeling [3], avoiding the use of expensive text-instruction labels in favor of visual MineCLIP embeddings. We apply unCLIP with classifier-free guidance [23] to create our agent called STEVE-1, which sets a new bar for open-ended instruction following in Minecraft with low-level controls (mouse and keyboard) and raw pixel inputs, far outperforming the baseline set by Baker et al. [5].

Our main contributions are as follows:

- We introduce a methodology, inspired by unCLIP [48], for instruction-tuning generative models of behavior without relying on a large dataset of expensive instruction labels.

- We apply this methodology to create STEVE-1, a Minecraft agent that can follow short-horizon open-ended text and visual instructions with a high degree of accuracy, all for only $60 of compute. We perform extensive evaluations of our agent, showing that it can robustly complete 12 of 13 goal-conditioned control tasks in our early-game evaluation suite in Minecraft. For long-horizon tasks[1] like crafting and building, we show that a basic version of prompt chaining can dramatically improve performance.

- We provide experimental evidence highlighting key factors for downstream performance, including pretraining, classifier-free guidance, data scaling, prompt-engineering, and other design choices. In particular, we show that unCLIP [48] and classifier-free guidance [23] translate well to sequential decision making and are essential for strong performance.

- We release model weights for STEVE-1 as well as training scripts and evaluation code to help foster more research into instructable, open-ended sequential decision-making agents.[2]

---

[1]Short-horizon tasks require few steps: e.g., go to a tree and chop it down, dig a hole. Long-horizon tasks take many steps: e.g., craft complex recipes from scratch, build a house.

[2]Model weights, training code, videos, and an interactive demo script are hosted on our project webpage at https://sites.google.com/view/steve-1.

## 2 Related Work

**Minecraft as a Test-bed for AI** Minecraft has gained popularity as a benchmark for AI research due to its complex and dynamic environment, making it a rich test-bed for reinforcement learning and other AI methods (e.g., [26, 19, 17, 21, 40, 62, 38, 9]). We leverage the MineRL environment [19] to research the creation of agents that can follow open-ended instructions in complex visual environments using only low-level actions (mouse and keyboard). We build STEVE-1 on top of two recent foundation models. In order to align text and videos, we use MineCLIP [17], a CLIP [47] model trained on paired web videos of Minecraft gameplay and associated captions. To train STEVE-1's policy, we fine-tune VPT [5], a foundation model of Minecraft behavior that is pretrained on 70k hours of web videos of Minecraft along with estimated mouse and keyboard actions. Several prior works [61, 62] have explored the use of LLMs in creating instructable Minecraft agents. These works typically use LLMs to make high-level plans that are then executed by lower-level RL [40, 62] or scripted [46] policies. Since STEVE-1 is a far more flexible low-level policy, the combination of STEVE-1 with LLMs is a promising direction for future work. Fan et al. [17] introduced an agent trained using RL with MineCLIP as a shaping reward on 12 different tasks and conditioned on MineCLIP-embedded text-prompts. However, this agent failed to generalize beyond the original set of tasks without further RL finetuning using the MineCLIP reward function. Cai et al. [9] proposed a Goal-Sensitive Backbone architecture for goal-conditioned control in Minecraft which is trained on a fixed set of goals, while STEVE-1 learns goal-reaching behavior from a large dataset in a self-supervised way without training on an explicit set of tasks.

**Foundation Models for Sequential Decision-Making** Foundation models which are pretrained on vast amounts of data and then finetuned for specific tasks have recently shown great promise in a variety of domains including language [8, 14, 59], vision [48, 10, 47], and robotics [7, 53, 25, 39, 65]. GATO [49] and RT-1 [7] have demonstrated the potential of training transformers to perform both simulated and real-world robotic tasks. With the exception of Kumar et al. [30], which uses Q-learning, the vast majority of cases [32, 7, 49] where deep learning has been scaled to large, multitask offline-RL datasets have used supervised RL. Supervised RL (e.g., [42, 18, 12]) works by framing the sequential decision-making problem as a prediction problem, where the model is trained to predict the next action conditioned on some future outcome. While these approaches are simple and scale well with large amounts of compute and data, more work is needed to understand the trade-offs between supervised RL and Q-learning or policy gradient-based methods [43, 44, 6, 55]. Recent works explore the use of hindsight relabeling [3] using vision-language models [47, 2] to produce natural language relabeling instructions. DIAL [65] finetunes CLIP [47] on human-labeled trajectories, which is then used to select a hindsight instruction from a candidate set. Sumers et al. [56] uses Flamingo [2] zero-shot for hindsight relabeling by framing it as a visual-question answering (VQA) task. In contrast, STEVE-1 relabels goals using future trajectory segment embeddings given by the MineCLIP [17] visual embedding.

**Text-Conditioned Generative Models** There has been a recent explosion of interest in text-to-X models, including text-to-image (e.g., [48, 51, 50]), text-to-3D (e.g., [27, 35]), and even text-to-music (e.g., [1]). These models are typically either autoregressive transformers modeling sequences of discrete tokens [60, 8] or diffusion models [24]. Most related to our work is unCLIP, the method used for DALL•E 2 [48]. unCLIP works by training a generative diffusion model to sample images from CLIP [47] embeddings of those images. By combining this model with a prior that translates text to visual CLIP embeddings, unCLIP can produce photorealistic images for arbitrary text prompts. unCLIP and many other diffusion-based approaches utilize a technique called classifier-free guidance [23], which lets the model trade-off between mode-coverage and sample fidelity post-training. We utilize the basic procedure of unCLIP and classifier-free guidance for training STEVE-1.

## 3 Method

Inspired by the rapid recent progress in instruction-tuning Large Language Models (LLMs), we choose to leverage the recently released Video Pretraining (VPT) [5] model as a starting point for our agent. Since VPT was trained on 70k hours of Minecraft gameplay, the agent already has vast knowledge about the Minecraft environment. However, just as the massive potential of LLMs was unlocked by aligning them to follow instructions, it is likely that the VPT model has the potential

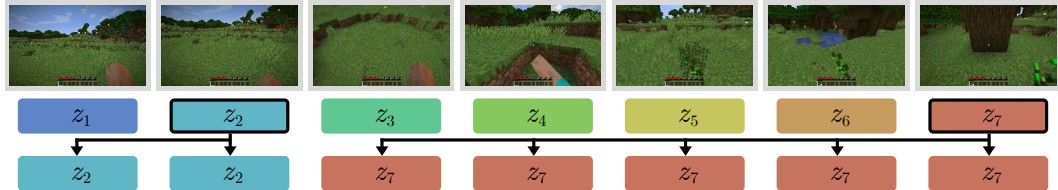

Figure 2: To create goal-conditioned data for finetuning, we randomly select timesteps from episodes and use hindsight relabeling to set the intermediate goals for the trajectory segments to those visual MineCLIP embeddings. This self-supervised data teaches the agent which actions lead to which states.

for general, controllable behavior if it is finetuned to follow instructions. In this work, we present a method for finetuning VPT to follow natural, open-ended textual and visual instructions, which opens the door for a wide range of uses for VPT in Minecraft.

Our approach is inspired by unCLIP, the method behind the recent text-to-image generation model, DALL•E 2 [48]. Our goal is to create a generative model of behavior in Minecraft conditioned on text instructions $y$. To do so, we utilize a dataset of Minecraft trajectory segments, some of which contain instruction labels $y$: $[(\tau_1, y_1), (\tau_2, y_2), \ldots, (\tau_n, \emptyset)]$ where $\tau$ is a trajectory of observations and actions. We also employ a pretrained CLIP model called MineCLIP [17], which generates aligned latent variables $z_{\tau_{t:t+16}}, z_y$, where $z_{\tau_{t:t+16}}$ is an embedding of any 16 consecutive timesteps from the trajectory. MineCLIP is trained using a contrastive objective on pairs of Minecraft videos and transcripts from the web. For simplicity of notation, we refer to the MineCLIP embedding of the last 16 timesteps of a trajectory segment as $z_{\tau_{\text{goal}}}$. Like unCLIP [48], we utilize a hierarchical model consisting of a prior and a policy:

- A *prior* $p(z_{\tau_{\text{goal}}}|y)$ that produces a latent variable $z_{\tau_{\text{goal}}}$ conditioned on a text instruction $y$.
- A *policy* $p(\tau|z_{\tau_{\text{goal}}})$ that produces a trajectory conditioned on a latent variable $z_{\tau_{\text{goal}}}$.

These two models can then be combined to produce a generative model of behaviors conditioned on text instructions:

$$p(\tau|y) = p(\tau, z_{\tau_{\text{goal}}}|y) = p(z_{\tau_{\text{goal}}}|y)p(\tau|z_{\tau_{\text{goal}}}) \tag{3.1}$$

### 3.1 Policy

To learn our policy, we finetune VPT, a foundation model of Minecraft behaviors $p_\theta(\tau)$ trained on 70k hours of Minecraft gameplay videos. Specifically, VPT consists of a ResNet [22] that processes frames of dimension $128 \times 128 \times 3$, and a Transformer-XL [15] which processes the frame representations and autoregressively predicts the next action using the joint hierarchical action space described in Baker et al. [5]. In order to modify the architecture to condition on goal information, we add an affine transformation of $z_{\tau_{\text{goal}}}$ to the output of the ResNet before passing it to the transformer:

$$\text{Process Frames:} \quad \text{ResNet}_\theta(o_t) \rightarrow x_t$$
$$[\text{+ Conditioning on MineCLIP Embedding Goal}]: \quad x_t \rightarrow x_t + W_\theta z_{\tau_{\text{goal}}} + b_\theta$$
$$\text{Predict Actions:} \quad \text{TransformerXL}_\theta(x_t, \ldots, x_{t+T}) \rightarrow a_{t+T}$$

In order to finetune VPT to condition on goals, we finetune the model using a method inspired by supervised RL approaches like Decision Transformer [12], GLAMOR [42], and GCSL [18]. We use a modification of hindsight relabeling which we call **packed hindsight relabeling** (see Figure 2) to generate a new dataset of trajectories with goals pulled from future states that periodically switch. In contrast with hindsight relabeling, packed hindsight relabeling packs multiple relabeled sequences into a single sequence. Specifically, our method to generate this dataset consists of two steps:

1. Given a trajectory $\tau$ with $T$ timesteps, randomly generate indices to select goals from: $i_1, i_2, \ldots, i_n$. These indices are chosen by starting at the first timestep and repeatedly sampling a new timestep by adding a random value to the previous timestep. This ensures that the data reflects that some goals may take longer to achieve than others.

2. For each chosen goal at timestep $i_j$, set the goals for timesteps $i_{j-1} + 1, \ldots, i_j$ to be the goal at timestep $i_j$, denoted $z_{\tau_{i_j}}$.

Our final dataset $\mathcal{D}_{\text{relabeled}}$ consists of observation sequences $(o_1, \ldots, o_T)$, action sequences $(a_1, \ldots, a_T)$, and packed hindsight relabeled goals $(z_1, \ldots, z_T)$. We then finetune VPT on this dataset using a supervised loss to predict each action autoregressively using a causal attention mask:

$$\mathcal{L}_{\text{policy}}(\theta) = \mathbb{E}_{\mathcal{D}_{\text{relabeled}}}[-\log p_\theta(a_t | o_{1\ldots t}, z_{1\ldots t})] \tag{3.2}$$

## 3.2 Prior

In order to condition not only on embeddings of visual goals but on latent goals, we need the prior, a model that produces a latent variable $z_{\tau_{\text{goal}}}$ conditioned on a text instruction $y$. Our model is a simple conditional variational autoencoder (CVAE) [54, 29] with a Gaussian prior and a Gaussian posterior. Rather than learn to condition directly on text, we choose to condition on frozen text representations from MineCLIP $z_y$. Thus, the prior learns a function to translate from a text embedding $z_y$ to a visual embedding $z_{\tau_{\text{goal}}}$ (see Appendix C.5). Both the encoder and decoder of our CVAE are parameterized as two-layer MLPs with 512 hidden units and layer normalization [4]. We train the model on our dataset, for which we have text labels $\mathcal{D}_{\text{labels}}$ using the following loss:

$$\mathcal{L}_{\text{prior}}(\phi) = \mathbb{E}_{(z_{\tau_{\text{goal}}}, z_y) \sim \mathcal{D}_{\text{labels}}}\Big[\text{KL}(q_\phi(z_{\tau_{\text{goal}}}|z_y)\|p(z_{\tau_{\text{goal}}})) - \mathbb{E}_{c \sim q_\phi(z_{\tau_{\text{goal}}}|z_y)}\big[\log p_\phi(z_{\tau_{\text{goal}}}|c, z_y)\big]\Big] \tag{3.3}$$

## 3.3 Datasets

To train our policy, we gather a gameplay dataset with 54M frames ($\approx$ 1 month at 20FPS) of Minecraft gameplay along with associated actions from two sources: contractor gameplay and VPT-generated gameplay. To train our prior, we use a small dataset of text-video pairs gathered by humans and augmented using the OpenAI API `gpt-3.5-turbo` model [41] and MineCLIP. See Appendix D for more detailed dataset information.

**OpenAI Contractor Dataset**   We use 39M frames sourced from the contractor dataset which VPT [5] used to train its inverse dynamics model and finetune its policy. The dataset was gathered by hiring human contractors to play Minecraft and complete tasks such as house building or obtaining a diamond pickaxe. During gameplay, keypresses and mouse movements are recorded. We use the same preprocessing as VPT, including filtering out null actions.

**VPT-Generated Dataset**   We generate an additional dataset of 15M frames by generating random trajectories using the various pretrained VPT agents. The diversity of this dataset is improved by randomly switching between models during trajectories [44], randomly resetting the agent's memory, and randomly turning the agent to face a new direction.

**Text-Video Pair Dataset**   To train our prior model, which learns a mapping between text embeddings and visual embeddings, we also manually gather a small dataset of 2,000 text instructions paired with 16-frame video segments (less than a second) from our gameplay dataset. This dataset corresponds to less than 30 minutes of gameplay and takes just a few hours to collect. We augment this dataset by using the alignment between text and video embeddings from MineCLIP. For each text instruction, we find the top $k$ most similar gameplay segments in our dataset and use the corresponding 16-frame segment as additional training data. For augmentation, we also add 8,000 text-instructions generated by the OpenAI API `gpt-3.5-turbo` model [41], in addition to our 2,000 hand-labeled instructions.

## 3.4 Inference

At inference time, we use the prior to sample a latent goal $z_{\tau_{\text{goal}}}$ from the text instruction $y$. We then use the policy to autoregressively sample actions $a_t$ conditioned on the observation history $o_{1\ldots t}$ and the latent goal $z_{\tau_{\text{goal}}}$. Similar to the observation in Appendix I of Baker et al. [5], even with conditioning the policy often fails to follow its instruction and simply acts according to its prior behavior. To mitigate this, we borrow another trick used in image generation models: classifier-free guidance. Specifically, during inference we simultaneously compute logits for the policy conditioned on the goal $f(o_t, \ldots, o_{t+1}, z_{\tau_{\text{goal}}})$ and for the unconditional policy $f(o_t, \ldots, o_{t+1})$. We then compute a combination of the two logits using a $\lambda$ parameter to trade-off between the two:

$$\text{logits} = (1 + \lambda) \underbrace{f_\theta(o_t, \ldots, o_{t+1}, z_{\tau_{\text{goal}}})}_{\text{conditional logits}} - \lambda \underbrace{f_\theta(o_t, \ldots, o_{t+1})}_{\text{unconditional logits}} \quad (3.4)$$

By setting a higher value of $\lambda$, we can encourage the policy to follow actions that are more likely when conditioned on the goal and, as demonstrated in Section 4.5, this significantly improves performance. Also, in order to train the policy to generate these unconditional logits, we occasionally dropout the goal embedding $z_{\tau_{\text{goal}}}$ from the policy's input (with probability 0.1). This lets us generate both the conditional and unconditional logits using the same model with batch processing at inference time.

### 3.5 Evaluation

Evaluating the performance of our agent is a challenging task due to the wide variety of instructions that are possible and the difficulty of evaluating whether the agent has successfully achieved its task. We use a combination of programmatic evaluation metrics and automatic MineCLIP evaluation metrics to get a sense of the agent's capability level. We collectively refer to all of our evaluation tasks including the 11 evaluation tasks from Figure 3 and the two prompt chaining tasks from Section 4.3 as our *early-game evaluation suite*.

**Programmatic Evaluation**  We compute programmatic evaluation metrics by monitoring the MineRL [19] environment state throughout each evaluation episode. As done in VPT [5], we compute multiple programmatic metrics including travel distance and early-game item collection. The travel distance is the maximum displacement of the agent along on the horizontal (X-Z) plane, measured from the initial spawn point. For early-game inventory counts, we store the maximum number of log, seed, and dirt items seen in the agent's inventory during the episode.

**MineCLIP Evaluation**  We explore the use of text-visual alignment in MineCLIP latent space between trajectories and text or visual goals to evaluate our agent over a wider variety of tasks where programmatic evaluation isn't practical. To determine the degree to which a task has been completed at all during an evaluation episode, we record the minimum cosine distance between the (text or visual) goal embedding and the visual MineCLIP embedding at any timestep during an episode.

## 4 Results

In our experiments, we aim to answer the following questions:

1. How well does STEVE-1 perform at achieving both text and visual goals in Minecraft?
2. How does our method scale with more data?
3. What choices are important for the performance of our method?

### 4.1 Training Setup

We base our implementation off of the official VPT codebase[3]. The main STEVE-1 is trained using Pytorch [45] distributed data parallel on four A40 GPUs for 160M frames, or just under three epochs of our gameplay dataset. Hyperparameters are selected to match those in Baker et al. [5] with the exception of learning rate, which we set to 4e-5. Our models are optimized using AdamW [37]. See Table 1 for a full list of hyperparameters.

### 4.2 Performance on Textual and Visual Goals

Due to the hierarchical nature of our model, we can evaluate the performance of our agent at achieving either text or visual goals simply by choosing whether to use the prior to condition on text or bypass the prior and condition on a MineCLIP video embedding directly. We first tested our model on a set of 11 tasks that are achievable within the first 2.5 minutes of gameplay and which do not require multiple steps to complete (e.g., chop a tree or dig a hole, but not build a house). A complete list of the tasks and prompts we used for evaluation can be found in Table 3 in the appendix. To select visual goals for testing each of the evaluation tasks, we implemented a tool that searches through

---

[3]https://github.com/openai/Video-Pre-Training

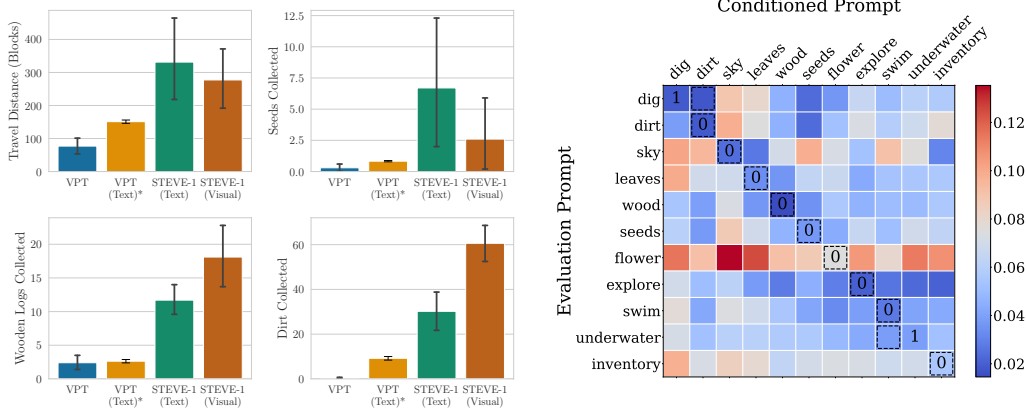

(a) Programmatic Evaluation          (b) MineCLIP Evaluation

Figure 3: **Left:** In our programmatic evaluations, STEVE-1 performed far better than the unconditional VPT agent `early-game-2x` and the text-conditioned VPT agent when prompted appropriately. The asterisk * in the "VPT (Text)*" indicates that this result was taken from Appendix I in [5], which had twice the episode length compared to our setting. On some tasks, visual outperforms text-based prompting, creating a gap that can likely be bridged through better prompt engineering. **Right:** Across our 11 MineCLIP evaluation tasks, STEVE-1 achieves the shortest distance between the episode and the MineCLIP goal embedding when prompted appropriately except for in two cases, where it mixes up digging and dirt and swimming and going underwater. This shows the strong general performance of STEVE-1 across a wide variety of short-horizon tasks. The dashed box marks minimum element along the row, and the diagonal number signifies the diagonal element's rank (0 means it is the minimum row element). See Figure 14 for sample frames from each of the 11 visual goals and Figure 13 for a success-rate version of this matrix.

10% of our gameplay dataset by finding the closest 16-frame videos to a given text prompt. We then manually selected a 16-frame video that clearly demonstrates the task being completed and use the corresponding MineCLIP video embedding as the goal embedding for that task. Screenshots of these visual goals can be found in Figure 14 in the appendix.

In Figure 3, we compare the performance of our text and visual-conditioned agents with the unconditional VPT agent and text-conditioned VPT agent (Appendix I in [5]) across our programmatic tasks. We find that when given the relevant text instruction, STEVE-1 collects $75\times$ more dirt, $4.9\times$ more wood, $22\times$ more seeds, and travels $4.3\times$ farther than the unconditional agent, and STEVE-1 collects $3.3\times$ more dirt, $4.4\times$ more wood, $8.1\times$ more seeds, and travels $2.2\times$ farther than the text-conditioned VPT agent. This represents a significant improvement over the reported performance of text-conditioned VPT, which collects several times fewer resources despite having twice as long of an episode to do so. We also run an automatic evaluation using MineCLIP embedding distances by measuring the minimum distance of a goal embedding to any frame in the episode. As shown in Figure 3b, the distance between the goal and the episode is significantly lower when the agent is conditioned on the corresponding visual goal than otherwise. Full results for STEVE-1 with both text and visual goals can be found in Appendix F.

In addition to our evaluations of STEVE-1, we also recorded several sample interactive sessions we had with the agent (controlling it in real-time by giving it written text instructions or specific visual goals). These sessions demonstrate STEVE-1's ability to responsively follow instructions in real-time in a variety of situations. We believe that such use-cases, where humans give an agent natural instructions that it can follow to complete tasks, will become increasingly important and have practical uses in the creation of instructable assistants and virtual-world characters. These videos, as well as videos of our agent performing our evaluation tasks, can be found at https://sites.google.com/view/steve-1.

### 4.3 Prompt Chaining

We also experiment with longer horizon tasks that require multiple steps, such as crafting and building. We explore two different prompting methods: directly prompting with the target goal, and a simple form of prompt chaining [11, 64, 16] where the task is decomposed into several subtasks and the

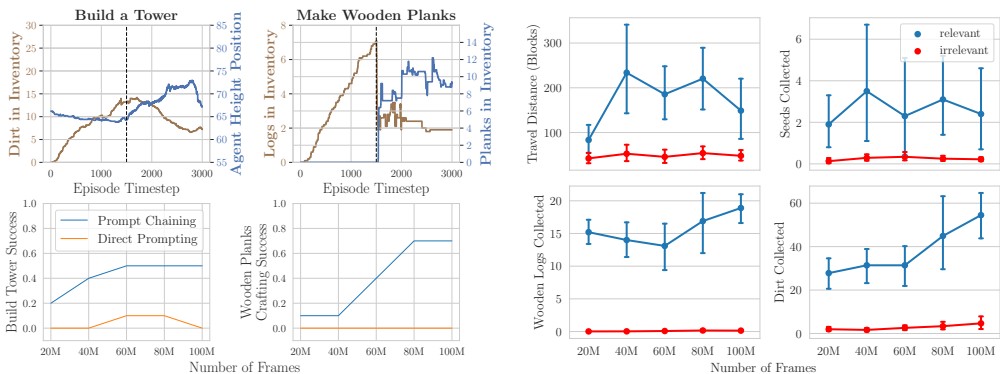

Figure 4: **Top left**: By sequentially chaining visual prompts like "get dirt" and "build a tower", STEVE-1 successfully gathers dirt and then uses this dirt to build a tower. The prompts switch at the dotted vertical line. **Bottom left**: The success rates of the chained prompts improve steadily as we train STEVE-1 on more data. **Right**: The performance of STEVE-1 in different tasks scales in different ways when conditioning on a relevant visual prompt for the metric versus other irrelevant visual prompts (e.g., the break wood prompt is the relevant prompt for the "Wooden Logs Collected" metric, while the other prompts are "irrelevant"). For instance, in the wood-collection and dirt-collection tasks, performance starts increasing after training on 60M frames of gameplay. See Figure 14 for sample frames from each visual prompt.

prompts are given sequentially for a fixed number of steps. We explore prompt chaining with visual goals for two tasks: 1) building a tower and 2) making wooden planks. When using prompt chaining, we first prompt STEVE-1 to gather dirt before building a tower, and to gather wooden logs before crafting wooden planks. Figure 4 shows that directly prompting STEVE-1 with the final tasks results in near-zero success rates. However, prompt chaining allows STEVE-1 to build a tower 50% of the time and craft wooden planks 70% of the time. For the tower building task, STEVE-1 immediately starts collecting dirt until the prompt switches, at which point its average height starts increasing rapidly and its dirt decreases as it builds a tower. Similarly, for the crafting wooden planks task, STEVE-1 immediately starts collecting a large amount of wooden logs until the prompt switches and it rapidly converts these wooden logs into wooden planks (causing the amount of wooden logs in its inventory to immediately decrease and the number of wooden planks to increase as it crafts more). Figure 4 visualizes the average item counts and agent height for the prompt chaining episodes. See Figure 18 and Figure 19 in the appendix for visualizations of specific prompt chaining episodes.

## 4.4 Scaling

Recent works in language modeling have found that scaling up pretraining FLOPs, by training on more data or by training a model with more parameters, can improve performance on downstream tasks [28, 57, 63]. In certain cases when measuring performance with metrics such as exact-match [52], performance improvement may appear to be "emergent" [63], appearing suddenly as the model is trained with more compute. Here, we aim to gain a basic understanding of how the performance of STEVE-1 on various tasks scales by training with more data (learning rate schedule is chosen appropriately).

To assess performance gain, we first isolated the performance of the policy from the prior, measuring performance of the agent through training on programmatic tasks (travel distance, seeds, logs, dirt) with visual goals. Due to compute constraints, we chose to use the 2x VPT model, which has 248M parameters. We found that both seed collection and travel distance did not improve significantly past 20M frames. From inspecting gameplay, we suspect that travel distance is a relatively easy task since it is close to VPT's default behavior of running around and exploring. For seed collection, performance remains suboptimal, suggesting that further scaling may be beneficial. This hypothesis is supported by the observation that performance on log and dirt collection remained roughly level until 60M frames when it began to rapidly improve. Figure 4 shows the scaling curves for STEVE-1 on each programmatic task when conditioning on relevant vs. irrelevant visual prompts for that task. Since we do not observe regression on any tasks as we train the model with more compute, we expect the model to continue to perform better as we train larger models on larger datasets.

We also evaluated the scaling properties of STEVE-1 for our multi-step tasks with and without prompt chaining. Without prompt chaining, the tasks remain challenging for STEVE-1 throughout training. However, we note that after 60M frames, STEVE-1 learns to gather wooden logs and build a small tower when told to build a tower. This is likely because our visual prompt for tower building shows a video of a tower being built out of wooden logs. With prompt chaining, the performance of STEVE-1 steadily increases with more data. We conjecture that this is because the success of a chained prompt requires the success of each element in the chain. Since different abilities emerge at different scales, one would expect chained prompts to steadily get more reliable as these subgoals become more reliably completed. In the case of building wooden planks, we note that crafting is one such task that gets significantly more reliable as the agent is trained on more data. Figure 4 shows the scaling curves for STEVE-1 on the prompt chaining tasks.

In summary, we see evidence of tasks that do not require much data for STEVE-1 to learn, tasks that steadily get more reliable as the agent is trained longer, and tasks where capability suddenly spikes after the agent reaches some threshold. Put together, this suggests that further scaling would likely significantly improve the agent, although we leave the task of predicting exactly how much performance there is to gain to future studies.

## 4.5 What Matters for Downstream Performance?

**Pretraining** Baker et al. [5] finds that by pretraining a behavioral prior with imitation learning on internet-scale datasets for Minecraft, the learned policy can be effectively finetuned to accomplish tasks that are impossible without pretraining. In this section, we demonstrate that pretraining is also massively beneficial for instruction-tuning in Minecraft. We hypothesize that due to the strong performance of STEVE-1 and the relatively small amount of compute ($\approx 1\%$ additional compute) used for instruction finetuning, most of the capabilities of our agent come from the pretraining rather than the finetuning. To test this hypothesis, we finetune several varients of STEVE-1 from various pretrained weights: `foundation-2x`, `bc-early-game-2x`, `rl-from-foundation-2x`, and with randomly initialized weights. In this experiment, each model was finetuned on 100M frames.

Figure 5 shows the performance of these models on our programmatic tasks with visual goals. Note that while an agent trained on our dataset from scratch can accomplish basic tasks like dirt collection fairly well, it is unable to find and chop down trees, in contrast to the pretrained agents. This demonstrates that the abilities present in the agent due to pretraining are successfully transferred to the finetuned agent. Out of all the pretrained weights we tried, we noticed that `rl-from-foundation-2x` performed the best, having qualitatively better performance at tasks like crafting and chopping down trees. Indeed, Figure 5 shows that this model has strong performance, likely due to the massive amount of compute it was trained with during its RL training [5].

**Classifier-Free Guidance** Baker et al. [5] observed that when conditioning the agent on text, it tended to ignore its instruction and instead perform the prior behavior learned during pretraining. As discussed in section 3.4, classifier-free guidance gives a knob for trading off between goal-conditioned and prior behaviors. Figure 5 shows the effect of this parameter $\lambda$ on the log and dirt collection tasks. The performance of the agent reaches its maximum around $\lambda = 5.0$ to $\lambda = 7.0$, after which it starts to drop off. These results demonstrate the importance of classifier-free guidance, which improves the performance of STEVE-1 by orders of magnitude.

**Prompt Engineering** Prompt engineering as a discipline has rapidly emerged over the last year due to the observation that the quality of the output of text-to-X models can dramatically change depending on the prompt [67]. For example, Table 5 in the appendix shows how a prompt for Stable Diffusion [50] might be written. By listing out the various attributes of the image such as visual medium, style, and the phrase "trending on ArtStation", the user is able to get a higher quality image [20, 36]. In this section, we explore how this same style of prompt engineering can improve the performance of STEVE-1. Figure 6 shows how a simple prompt of "get dirt" might be changed in order to more accurately specify the type of behavior that is desired. Just like in image generation models, the performance of STEVE-1 significantly improves by modifying the prompt in this fashion. By changing to more complicated prompts, STEVE-1 is able to collect $1.6\times$ more wood, $2\times$ more dirt, and $3.3\times$ more seeds.

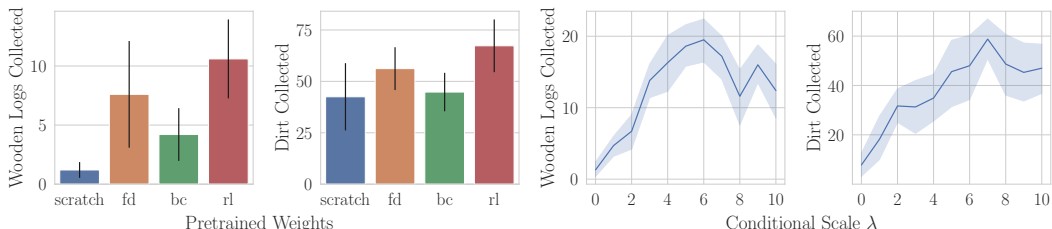

Figure 5: **Left:** We trained STEVE-1 on 100M frames starting from four different pretrained weights: random initialization (scratch), `foundation-2x` (fd), `bc-early-game-2x` (bc), and `rl-from-foundation-2x` (rl). The `rl-from-foundation-2x` agent is generally the most performant after fine-tuning. Using pretrained weights performs better than training from scratch, especially for more complicated tasks like collecting wood. **Right:** By using classifier-free guidance [23], STEVE-1 collects $7.5\times$ more dirt and $15\times$ more wood than when $\lambda = 0$ (no guidance). See Figure 17 in the Appendix for similar results with other programmatic tasks.

| Prompt | Dirt Collected |
|---|---|
| "break a flower" | 0.7 (-0.2, 1.6) |
| "collect seeds" | 2.7 (0.9, 4.5) |
| "dig as far as possible" | 3.9 (2.8, 5.0) |
| "get dirt" | 9.2 (5.7, 12.7) |
| "get dirt, dig hole, dig dirt, gather a ton of dirt, collect dirt" | **26.7 (19.9, 33.5)** |

Figure 6: Similar to in image generation, switching to a longer, more specific prompt dramatically improves the performance of STEVE-1. Values in parentheses are 95% confidence intervals.

## 5 Limitations and Conclusion

In this paper, we present a methodology for creating instruction-following foundation models of behavior. Specifically, by leveraging two existing pretrained foundation models: a behavioral prior (VPT [5]) and a domain-specific CLIP model (MineCLIP [17]), we create a powerful Minecraft agent that can follow short-horizon open-ended text and visual instructions, all for only $60 of compute. The resulting foundation model, STEVE-1, sets a new bar for open-ended instruction following in Minecraft with low-level controls (mouse and keyboard) and raw pixel inputs, far outperforming previous baselines and robustly completing 12 of 13 tasks in our early-game evaluation suite. We note that generalist agents such as STEVE-1 can have potential negative effects on society. We include a thorough discussion of these issues in Appendix A.

STEVE-1 is a significant advancement in creating generative models of text-to-behavior, but it has several limitations, as described in Appendix B. First, STEVE-1 is mostly proficient at achieving short-horizon tasks while struggling with longer-horizon tasks. While prompt chaining is a promising approach for improving performance on complex tasks, more can be done in future work to improve performance. Another limitation we observe is that prompt engineering, as with other generative models, can be unintuitive and time-consuming. Future work should investigate improving the steerability of STEVE-1 through a better understanding of natural language prompts. Additionally, we note that evaluating and describing the capabilities of open-ended generalist agents is an open research problem itself since capability depends strongly on preconditions, prompt engineering, and our own ability to come up with varied and challenging tasks. Finally, since our approach is not specific to the Minecraft domain, we hope that the method used to create STEVE-1 can inspire future work in creating powerful generalist agents in other domains and environments.

## Acknowledgements

All of the authors gratefully acknowledge funding for this research from the Natural Sciences and Engineering Research Council of Canada (NSERC) and the Canada CIFAR AI Chairs Program (Vector Institute for Artificial Intelligence). SL is supported by a Vector Institute internship and by an NSERC Discovery Grant. KP is supported by an NSERC PGS-D award. HC is supported by an NSERC CGS-D award. JB acknowledges funding from the Canada CIFAR AI Chairs program, Fujitsu Japan, and an Amazon Research Award. In addition to NSERC and CIFAR (Vector Institute), SM acknowledges funding from Microsoft Research. We thank Silviu Pitis, Romi Lifshitz, Forest Yang, and Yongchao Zhou for their helpful comments; Alisa Wu and Ziming Chen for their contributions to the text-video pair dataset; and Finn Paster for the logo and graphic for the website. Resources used in preparing this research were provided, in part, by the Province of Ontario, the Government of Canada through CIFAR, and companies sponsoring the Vector Institute for Artificial Intelligence (`www.vectorinstitute.ai/partners`).

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

# A   Broader Impact

With the increasing capability level of artificial intelligence comes many potential benefits and also risks. On the positive side, we anticipate that the techniques that used to create STEVE-1 could be applied to the creation of helpful agents in other sequential decision making domains, including robotics, video games, and the web. Our demonstration of such a low cost approach to creating a powerful, instruction-following model also has the potential to improve the democratization of AI. However, on the negative side, agents pretrained on large internet datasets reflect the biases of the internet and, as suggested by our experiments, these pretraining biases can potentially remain after instruction-tuning. If not addressed carefully, this could lead to devastating consequences for society. We hope that while the stakes are low, works such as ours can improve access to safety research on instruction-following models in sequential decision-making domains.

# B   Limitations and Future Work

## B.1   Goal Misgeneralization

One of the most common mistakes that STEVE-1 makes during evaluation is to overgeneralize. For instance, if we prompt STEVE-1 with a video of someone punching a cow, it may simply run to the nearest animal and punch that animal instead. This is related to the concept of goal misgeneralization [31]. Generalization can be helpful when the task we assign the agent is impossible to achieve from the current state and the agent instead performs a closely related action, but harmful when the task is achievable. We note two things: first, we believe the powerful generalization ability of STEVE-1 probably comes from the MineCLIP embeddings and it especially improves the ability of STEVE-1 to follow visual instructions when the exact items or blocks nearby are not available in the current environment, which is an extremely common scenario. Second, we notice that the tendency of the agent to misgeneralize decreases with scale. For example, with a model trained on less data, we find that asking the agent to look up and punch a tree to get a wooden log often resulted in the agent looking in the air and punching nothing; training the model on more data results in the agent first walking over to a nearby tree and looking up to get a wooden log. Future work should look to measure more closely the relationship between misgeneralization and scale.

## B.2   Random Selection of Hindsight Goals

In this work, we choose to **randomly** select future episode timesteps as hindsight goals, rather than use a more sophisticated strategy. This is primarily due to the simplicity of the approach, but also to ensure a diverse and unbiased coverage of potential goals achievable within the short horizon. Future works can investigate the effects of alternative approaches that filter for semantically interesting timesteps as goals.

## B.3   Difference in Performance Between Text and Visual Goals

In our experiments, we observed that STEVE-1 often performed better when conditioned with visual goals compared to text goals converted through the prior. There are several potential factors that could contribute to this performance gap between text and visual conditioning. First, the prior model may not be accurately capturing the full meaning of the text prompt. Training the prior on more data or using a more powerful model architecture could potentially improve the quality of the sampled latent goals. Second, the visual goals can provide more precise demonstrations of the desired behavior. Text goals are inherently more ambiguous. Providing additional information such as the observation context to the prior and further prompt-engineering to make text prompts more detailed and less ambiguous, could help close this gap. Because text conditioning provides more flexibility and potential for generalization, closing the performance gap between text and visual conditioning is an important direction for future work.

## B.4   Challenges in Evaluating STEVE-1

See Table 4 for a non-exhaustive list of tasks that the STEVE-1 agent is able to achieve. It is worth mentioning that evaluating and describing the capabilities of open-ended generalist agents is an open

research problem itself since capability depends strongly on preconditions, prompt engineering, and our own ability to come up with varied and challenging tasks. For example, there are many recent works on the evaluation of LLMs (e.g., [34, 33, 66]) which highlight these challenges.

That being said, there are a number of tasks which STEVE-1 is unable to accomplish. As previously mentioned, long-horizon tasks such as obtaining a diamond or building a house are currently beyond the capability of STEVE-1. Further, STEVE-1 also struggles with more complex crafting tasks like crafting an enchanting table, bookcase, or boat. Again, the virtually limitless and open-ended nature of tasks in Minecraft makes it very difficult to test generalist agents in this domain. We hope future works develop more sophisticated methods to evaluate the performance of generalist agents on short and long-horizon tasks (potentially through an extension of our MineCLIP evaluation method).

It is also worth noting that it is currently not possible to test STEVE-1 or VPT [5] in the MineDojo [17] environment, which is meant for generalist agent evaluation, since the action spaces are not compatible. We believe that bridging this gap could be greatly beneficial to the generalist agent community and we hope future works investigate this further.

### B.5 Towards Improved Long-Horizon Performance

STEVE-1 is a significant advancement in creating generative models of text-to-behavior, but it has several limitations. First, STEVE-1 is mostly proficient at achieving short-horizon tasks while struggling on longer-horizon tasks like obtaining a diamond. Solving long-horizon tasks while taking actions using low-level mouse/keyboard controls is a very challenging and exciting research direction and, while prompt chaining is a promising approach for improving performance on complex tasks, more can be done in future work to improve performance.

One potential bottleneck is the fact that during packed hindsight relabeling, the hindsight goals are limited to at most 200 timesteps in the future ($\sim$10 seconds). Thus, tasks which require more than 200 timesteps to complete are technically out-of-distribution for STEVE-1. Although sampling hindsight goals from farther into the future could theoretically enhance long-horizon performance, our experiments in Appendix C.4 indicate that the performance tends to decrease if we increase this hyperparameter too much. We suspect that while increasing this hyperparameter may be able to improve long-horizon performance, it also increases noise and comes at the cost of reducing performance on short-horizon goals. Investigating whether it is possible to achieve a better tradeoff is an important avenue for future work. We also suspect that the long-horizon capabilities of STEVE-1 could be improved through scaling or finetuning with reinforcement learning, or leveraging LLMs or VLMs to automatically provide prompt chains to the STEVE-1 agent.

### B.6 Applying the STEVE-1 Approach to Other Domains

We designed STEVE-1 for Minecraft due to the availability of two key ingredients: (1) a strong behavioral prior (VPT [5]), and (2) a powerful visual-language model which maps text and video to a joint embedding space (MineCLIP [17]). However, the method used to create STEVE-1 is not specific to the Minecraft domain. Given the rapid development of generative models, we expect that similar models to VPT and MineCLIP will become available in many other domains. As these models become available, future work could investigate the applicability of the STEVE-1 approach to these other domains.

## C  Additional Ablations

In this section, we describe additional ablations on design choices for our method, including the use of classifier-free guidance during training, text augmentation strategies, different VAE variants, and varying chunk sizes during finetuning. We use programmatic evaluation metrics to compare the performance of the various ablations.

### C.1  Classifier-Free Guidance During Training

We examine the importance of using classifier-free guidance during training by finetuning a model with *no* guidance which does not drop out the goal embedding $z_{\tau_{goal}}$ from the policy's input (i.e., $p_{uncond} = 0.0$) and comparing it to the version which uses guidance ($p_{uncond} = 0.1$). The chunk

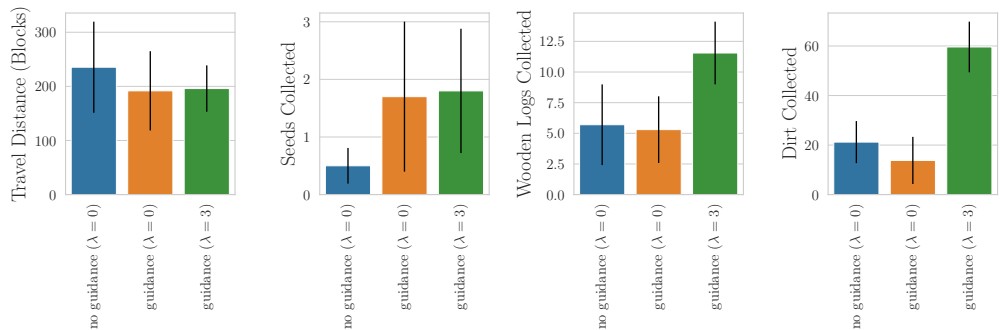

Figure 7: **Ablation on Guidance**. In the "*no guidance*" variant, we set $p_{uncond} = 0$, meaning that we do not drop any $z_{\tau_{goal}}$ from the policy's input during training. The "*guidance*" variants set $p_{uncond} = 0.1$, dropping 10% of the time during training. Whereas the "*no guidance*" model is only compatible with $\lambda = 0$ at inference, the "*guidance*" model can use $\lambda > 0$, allowing for better performance.

size is set to the range 15 to 50 and we train each policy for 100M frames. In Figure 7, we compare the performance of using *visual* goals (MineCLIP video embedding) on the no guidance model using conditional scale $\lambda = 0$ and the guidance model using conditional scales $\lambda = 0$ and $\lambda = 3$. We observe that while the no guidance model slightly outperforms the guidance model at $\lambda = 0$ across a few metrics, the agent with guidance outperforms the no guidance agent by a factor of 2 to 3 times for the inventory collection tasks when we increase the conditional scale to $\lambda = 3$ (which we cannot do for the no guidance model). For the travel distance metric, both of the guidance versions perform similarly to the no guidance version.

## C.2 Text Augmentation

During finetuning, instead of using only self-supervision with future MineCLIP video embedding as the goal, we considered using the *text* embeddings from the 2,000 human labeled trajectory segments as goal embeddings, either solely or in addition to the self-supervised video embeddings. In order to more fairly compare with the CVAE prior approach, we augment the human-labeled data with additional text-gameplay pairs generated as described in Appendix E.2. We implement this experiment by replacing the visual embeddings used for relabeling in Algorithm 1 with text embeddings, when available, with a 90% probability. To experiment with not using visual embeddings at all, we can replace the visual embeddings with zeros in the same way. In Figure 8, we observe that using only the visual embeddings during training, in combination with the CVAE, can outperform using MineCLIP text embeddings directly in the other two baselines. In this experiment, the chunk size is set to the range 15 to 50 and we train each policy for 100M frames.

## C.3 VAE Variants

We study the dataset used to train the CVAE prior model. In Figure 9, we observe that augmentation helps in some programmatic tasks, including the dirt and seed collection tasks, but slightly hurts the wooden log collection and travel distance metrics. In this experiment, we use the same policy with each CVAE variant and we tune the conditional scale $\lambda$ for each variant. The chunk size is set to the range 15 to 200 and we train the policy for 100M frames.

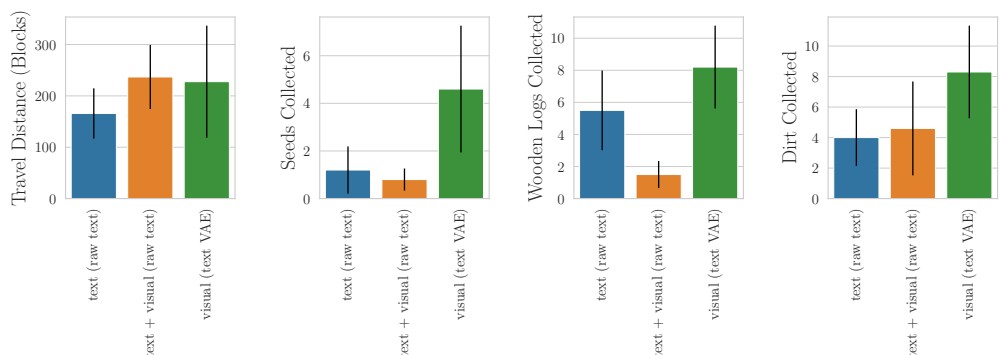

Figure 8: **Ablation on Text Augmentation**. In the "*text (raw text)*" ablation, we train the model using only the text labels from human labelled trajectory segments, and directly use the MineCLIP text embedding of the text label as the goal embedding during training and at inference. For the "*text + visual (raw text)*" ablation, we use both the visual embedding in self-supervised manner and the text embedding from the human labelled trajectory segments during training and use the MineCLIP text embedding during inference. Even with augmentation, the dataset only contained around 2% text embeddings. The "*visual (text VAE)*" version is as reported in the main method, using the CVAE to convert MineCLIP text embedding to visual embedding during inference.

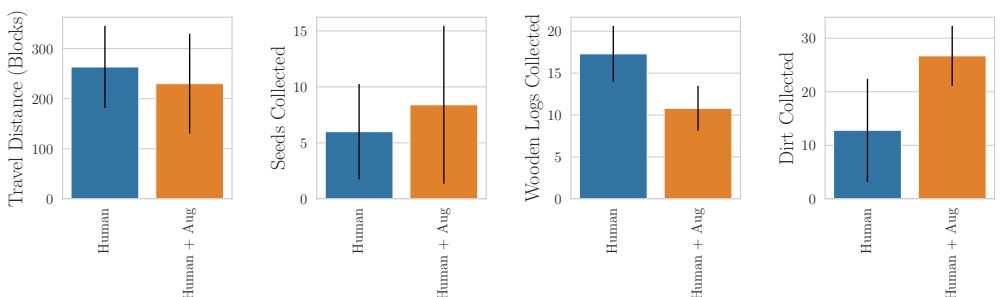

Figure 9: **Ablation on VAE Training Data**. "*Human*" baseline uses only the 2,000 human-labelled trajectory segments (text-video pairs), as training example for the CVAE prior model. "*Human + Aug*" baseline adds additional pairs of text-video examples as described in Section 3.3.

## C.4 Chunk Size

During finetuning, we compare different goal chunk sizes by varying the `max_btwn_goals=[100,200,300,400]`, while keeping the `min_btwn_goals=15`. See Algorithm 1 for more details. A larger `max_btwn_goals` introduces more noise, with actions that led to achieving the further away goal being less correlated to the actions present in that goal chunk. In Figure 10, we observe that the best `max_btwn_goals` chunk size is around 200, and increasing the chunk size beyond that causes a drop in performance. We train each policy for 160M frames and tune the conditional scale for each.

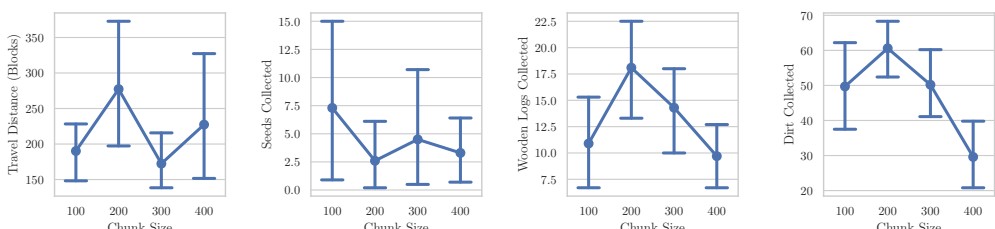

Figure 10: **Ablation on Segment Chunk Size**. We vary the `max_btwn_goals` parameter in Algorithm 1. The performance is roughly the best at around 200, beginning to decline with greater values.

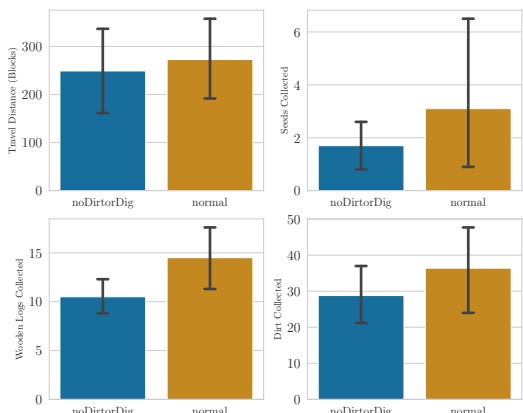

Figure 11: Even without training the prior on the concept of dirt or digging at all, STEVE-1 can still be instructed to dig holes and get dirt. This demonstrates that STEVE-1 can generalize to unseen text instructions.

## C.5 Generalization to Novel Text Instructions

We train the prior on a dataset of human and GPT-generated instructions designed to be representative of the tasks that appear in our gameplay dataset. Here, we have performed a set of simple generalization experiments to measure the degree to which the prior can generalize to unseen instructions.

**Instruction Training Set Contamination**   The "Text Prompt" column in Table 3 shows which of the prompts used in evaluation show up in our training dataset. Among our evaluation instructions, the bolded and italicized instructions show up in the instruction-trajectory dataset. While some instructions do show up, most of the instructions do not show up in our training set (verbatim).

**Training Set Decontamination**   To measure the effect on performance of removing a concept from the training set, we ran an experiment where we removed every instruction with the words "dirt" or "dig" in them and retrained the VAE model. This corresponds to around 10% of the instructions. As shown in Figure 11, we found that even without training on the concept of dirt or digging at all, STEVE-1 can still be instructed to dig holes and get dirt. This demonstrates clearly that STEVE-1 can generalize to unseen text instructions — likely because most of the text-understanding comes from the pretrained MineCLIP model which was trained on a highly diverse dataset of YouTube videos and captions. The prior VAE only needs to learn a mapping between the text and visual MineCLIP embeddings. Note that there is a slight decrease in performance across all tasks likely due to the smaller VAE training set ($\sim$ 10% less).

The instruction-following capability of STEVE-1 is shared between: the policy, which learns to follow instructions in the visual MineCLIP embedding space; the MineCLIP text-encoder, which is trained to align well with the visual embeddings and performs most of the text-understanding; and our prior VAE model, which learns a simple function to translate between text and visual embeddings. This modeling setup lets us fully exploit pretrained models such as MineCLIP to gain impressive language understanding without relying on having our own large datasets or compute.

## D   Dataset Details

### D.1   Gameplay Dataset

Our gameplay dataset consists of two types of episodes: 7,854 episodes (38.94M frames) of a contractor dataset made available from Baker et al. [5] and 2,267 episodes (14.96M frames) of gameplay generated by running various pretrained VPT agents.

**OpenAI Contractor Dataset**   The majority of our data comes from the contractor data used to train VPT [5]. OpenAI released five subsets of contractor data: 6.x, 7.x, 8.x, 9.x, and 10.x. We use an equal

mix of 8.x, 9.x, and 10.x, which correspond to "house building from scratch", "house building from random starting materials", and "obtain diamond pickaxe". Contractors were given anywhere from 10 to 20 minutes to accomplish these goals to the best of their abilities while their screen, mouse, and keyboard were recorded.

**VPT-Generated Dataset**   We generated additional data by generating episodes using various pre-trained VPT agents. In order to increase the diversity of data as well as to get data of the agent switching tasks randomly throughout the middle of episodes, we added random switching between the different pretrained agents during episodes. Specifically, at the beginning of an episode we randomly sample two VPT agents from (`foundation_model_2x`, `bc_early_game_2x`, `bc_house_3x`, `rl_from_foundation_2x`, `rl_from_house_2x`) and switch between them at each timestep with a probability of $1/1000$. Since the RL agents all act quite similarly, we avoid sampling two RL agents at once. Additionally, with a probability of $1/750$ each timestep, we cause the agent to spin a random number of degrees. This adds more data where the agent spontaneously changes tasks, increasing downstream steerability.

## D.2   Text-Video Pair Dataset

We gathered a small dataset of 2,000 human-labelled trajectory segments (text-video pairs) by manually labeling gameplay from our datasets. We used a simple web app that presented a video of 16 frames to the user from a randomly sampled episode. This only corresponds to 32,000 frames of labeled data, which corresponds to labeling 0.06% of the full dataset, or 27 minutes of labeled data. However, as discussed in Appendix E.2, combining this with automatically labeled data using `gpt-3.5-turbo` and MineCLIP results in a strong prior model.

## D.3   Prompt Design

In our experiments we used both short and longer prompts. The short prompts are either taken from previous literature (e.g., the language-conditioning experiment in the VPT appendix [5]) or they were simply the first prompt we tried. The longer prompts were created by taking inspiration from the prompt engineering methods used with text-to-image models such as Stable Diffusion [50]. To design these prompts, we simply strung together a lot of terms related to our task in order to increase the specificity of the prompts. We were excited to discover that this style of prompt design inspired by the prompt-engineering community works well in STEVE-1.

# E   Training Details

## E.1   Policy Training

STEVE-1 was trained using distributed data parallel in PyTorch [45]. During training, segments of 640 timesteps were sampled from the dataset. Due to memory constraints, these segments were further broken up into chunks of 64, which are processed sequentially. Since VPT uses a Transformer-XL [15], this sequential processing lets the policy attend to previous batches up to the limit of its context length. We optimized the weights using AdamW [37] with a maximum learning rate of 4e-5 and a linear warmup for the first 10M frames followed by a cosine learning rate decay schedule that decays to 10% of the original learning rate. See Table 1 for an exhaustive list of hyperparameters used during training.

During training, we sample data using packed hindsight relabeling (Figure 2). This involves sampling a segment of an episode, randomly selecting some timesteps at which to change goals, and then filling in the corresponding goal embeddings for the entire episode with the embeddings from the corresponding goal segments. See Algorithm 1 for a detailed explanantion of packed hindsight relabelling.

## E.2   Prior Training

The prior model is a simple CVAE [54] that conditions on MineCLIP [17] text embeddings and models the conditional distribution of visual embeddings given the corresponding text embedding. This model is trained on a combination of around 2,000 hand-labeled trajectory segments and augmented

| Hyperparameter Name | Value |
|---|---|
| `trunc_t` | 64 |
| `T` | 640 |
| `batch_size` | 12 |
| `num_workers` | 4 |
| `weight_decay` | 0.039428 |
| `n_frames` | 160M |
| `learning_rate` | 4e-5 |
| `optimizer` | AdamW [37] |
| `warmup_frames` | 10M |
| `p_uncond` | 0.1 |
| `min_btwn_goals` | 15 |
| `max_btwn_goals` | 200 |
| `vpt_architecture` | 2x |

Table 1: Policy Hyperparameters

| Hyperparameter Name | Value |
|---|---|
| `architecture` | MLP |
| `hidden_dim` | 512 |
| `latent_dim` | 512 |
| `hidden_layers` | 2 |
| `batch_size` | 256 |
| `learning_rate` | 1e-4 |
| $\beta$ | 0.001 |
| `n_epochs` | 50 |
| `n_search_episodes` | 2000 |
| `k` | 5 |
| `offset` | 8 |

Table 2: Prior Hyperparameters

---

**Algorithm 1:** Sampling Episode Segments with **Packed Hindsight Relabeling**

---

**Function** `sample_episode_segment`(*T, min_btwn_goals, max_btwn_goals*)

    segment = sampleSegment(episode, T)
    curr_timestep = segment.start
    goal_switching_indices = []
    **while** curr_timestep < segment.end **do**
        curr_timestep += uniform(min_btwn_goals, max_btwn_goals)
        goal_switching_indices.append(curr_timestep)

    relabeled_goal_embeds = []
    **for** n in range(1, len(goal_switching_indices)) **do**
        relabeled_goal_embeds[$i_{n-1}$:$i_n$] = segment.goal_embeddings[$i_n$]

    **return** segment.obs, segment.actions, relabeled_goal_embeds

---

with additional data by automatically searching for text-gameplay pairs from our gameplay dataset. This is done using the following steps:

1. Combine the 2,000 text labels with 8,000 additional labels generated by querying `gpt-3.5-turbo`.

2. For each of these 10,000 text labels, search through 1,000 episodes sampled from the gameplay dataset to find the top 5 closest visual MineCLIP embeddings to the text embedding of the text label.

These 50,000 automatically-mined text-video pairs are added to the original 2,000 hand-labeled examples to form the final dataset used for prior training.

We noticed when prompting STEVE-1 using visual goals that when the visual goal showed the agent hitting a block but not following through and breaking it that STEVE-1 actually avoided breaking blocks. Unfortunately, many of the automatically discovered text-gameplay clips include gameplay of this kind. In order to prevent this issue, we added an offset to the embeddings found in this manner. By selecting embeddings from a timestep `offset` steps after the originally-selected timestep, the agent is much more likely to follow through with breaking blocks.

We trained our prior model for 50 epochs on this dataset and used early-stopping with a small validation set. An exhaustive list of hyperparameters used for creating the prior model can be found at Table 2.

### E.3 Training Costs

The \$60 cost we report corresponds to the cost of renting a 8xA10g node using spot instances on AWS for 12 hours using our instances prices in May 2023.

## F Additional Visualizations

### F.1 MineCLIP Evaluation

We ran MineCLIP evaluation on both text and visual prompts. The MineCLIP evaluation results can be found in Figure 12.

### F.2 Steerability with Programmatic Metrics

Similar to Figure 20 in the VPT appendix [5], we plot the programmatic metric performances (mean and 95% confidence intervals) across the different goal prompt conditioning, both using visual prompts (Figure 15) and text prompts with CVAE prior (Figure 16) conditioning, on our policy trained with hyperparameters in Table 1 and using conditional scaling $\lambda = 7$ (for visual prompts) and $\lambda = 6.0$ (for text prompts with CVAE prior). Each conditioning variant is run with 10 trials, each trial with a different environmental seed and with an episode length of 3000 timesteps (2.5 minutes gameplay). Across the conditioning variant, we use the same set of environmental seeds. For comparison, we also plot the metrics for an unconditional VPT (`early_game`) agent ("*VPT (uncond)*") and the text-conditioned agent investigated in VPT appendix [5] ("*VPT (text)\**") when conditioned on the relevant text. When using *visual* goal conditioning, we the use MineCLIP video encoder to embed a 16-frame clip of the agent performing the desired task taken from our training dataset. An example frame from each of the visual goals is illustrated in Figure 14. When using *text VAE* goal conditioning, we use the MineCLIP text encoder to encode the text prompts (Table 3) and use the CVAE prior to sample the goal embedding from the MineCLIP text embedding.

We note several differences in our experimental setup compared to that in VPT [5]. We only run our evaluation episodes for 3000 timesteps, equivalent to 2.5 minutes of gameplay, compared to 5 minutes in the VPT paper. Due to a limited computational budget, we generate 10 episodes per conditioning variant, and 110 episodes for the unconditional ("*VPT (uncond)*"), compared to VPT's 1000 episodes. Lastly, when measuring the inventory count, we log the maximum inventory count seen throughout the episode, which is a lower bound on the potential number of items collected since the agent can later throw out, place, or use these items to craft. As a result of these caveats, we denote the "*VPT (text)\**" legend in Figure 15 and Figure 16 with an asterisk as we use the results reported in [5] directly for comparison.

We make several observations. First, we observe that our agents is more *steerable*: when conditioned to collect certain items (in bold), the agent collects (relatively) many more of those items than when conditioned on other instructions unrelated to that item, as well as compared to the unconditional VPT. When conditioned on tasks unrelated to the item (e.g. break a flower when interested in measuring logs collected), we also observe that the agent pursues that item *less* than the unconditional agent. Second, we observe that for the bolded instructions which we expect to stand out, we outperform VPT performance (dashed blue line) [5], even with half the amount of time in the episode rollout. This suggests that our agent is both more steerable relative to the unconditioned VPT agent and the text-conditioned VPT agent investigated in the VPT appendix [5].

### F.3 Prompt Chaining Visualization

We visualize two specific episodes from the prompt chaining experiments in Section 4.3 in Figure 18 (building a tower) and Figure 19 (crafting wooden planks).

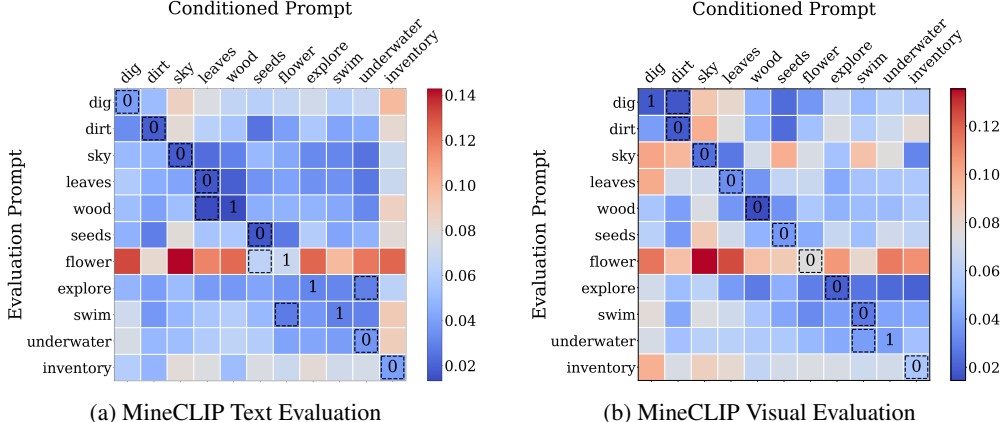

| (a) MineCLIP Text Evaluation | (b) MineCLIP Visual Evaluation |

Figure 12: **MineCLIP Evaluation**. We measure the cosine distance between the goal embedding given to the agent and the MineCLIP video embeddings throughout the episode and record the minimum across the episode. Dashed box indicates the minimum along the row, and the number in the diagonal box indicates the rank of the diagonal element in the row (0 specifies that the diagonal is the minimum element in the row). The ideal performance would be where the minimum values of each row lie on the diagonal. That is, the agent performs a specific task best when it is asked to perform that specific task. **Left:** We use the prior to convert the text into the goal embedding. Across our 11 text MineCLIP evaluation tasks, STEVE-1 achieves the shortest distance between the episode and the MineCLIP goal embedding when prompted appropriately for most cases. This shows the strong general performance of STEVE-1 across a wide variety of short-horizon tasks. **Right:** We embed the visual goal loops (Figure 14) with MineCLIP video encoder. Across our 11 visual MineCLIP evaluation tasks, STEVE-1 achieves the shortest distance between the episode and the MineCLIP goal embedding when prompted appropriately except for in two cases, where it mixes up digging and dirt and swimming and going underwater. This shows the strong general performance of STEVE-1 across a wide variety of short-horizon tasks.

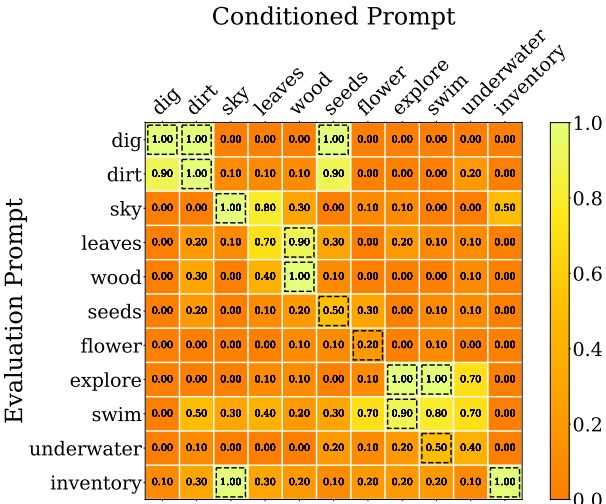

Figure 13: **Visual Evaluation Success-Rate Matrix.** We manually reviewed the same videos used for the MineCLIP Visual Evaluation matrix in Figure 3b in order to verify that the MineCLIP scores correspond well to human judgment. Thus, the values in this matrix are subject to human error and subjectivity. Each cell value shows how often the agent achieves the Evaluation Prompt when conditioned on the Conditioned Prompt (success-rate). The dotted cell(s) is/are the maximum value in the row. Across the tasks, STEVE-1 achieves the highest success-rate when prompted appropriately except in three cases, where it breaks wood more than leaves, explores more than it swims, and swims more than it goes underwater. This shows the strong general performance of STEVE-1 across a wide variety of early-game short-horizon tasks.

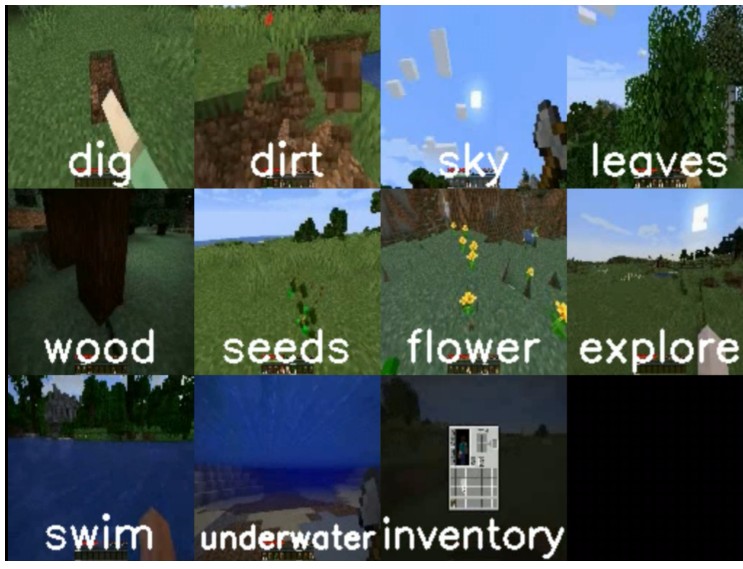

Figure 14: Sample frames from each of the 11 visual goals. Note that the text overlaid on the frame is *not* present when we encode the 16-frame clip with the MineCLIP video encoder, and is only present for the figure visualization.

| Shortened Name | Conditioning Variant Name | Text Prompt |
|---|---|---|
| dig | dig as far as possible | ***dig as far as possible*** |
| dirt | get dirt | get dirt |
| sky | look at the sky | look at the sky |
| leaves | break leaves | ***break leaves*** |
| wood | chop a tree | ***chop a tree*** |
| seeds | collect seeds | ***collect seeds*** |
| flower | break a flower | break a flower |
| explore | go explore | ***go explore*** |
| swim | go swimming | go swimming |
| underwater | go underwater | go underwater |
| inventory | open inventory | ***open inventory*** |
| dirt (engineered) | get dirt . . . | get dirt, dig hole, dig dirt, gather a ton of dirt, collect dirt |
| wood (engineered) | chop down the tree . . . | chop down the tree, gather wood, pick up wood, chop it down, break tree |
| seeds (engineered) | break tall grass . . . | break tall grass, break grass, collect seeds, punch the ground, run around in circles getting seeds from bushes |

Table 3: A summary of the different ways we refer to the 11 early-game evaluation task prompts. "Shortened Name" is the way we refer to the prompts in any success-rate matrix or MineCLIP matrix. This is also how we refer to the visual prompts which have no actual text. "Conditioning Variant Name" is the name used to refer to the "Text Prompts" in Figure 16, since not all text prompts fit in the figure. A "Conditioning Variant Name" with ". . ." indicates that this is an engineered text prompt that does not fit in the figure. Also, in reference to the experiments Appendix C.5, the bolded and italicized prompts in the "Text Prompt" column are those that were present verbatim in the text-video pair dataset.

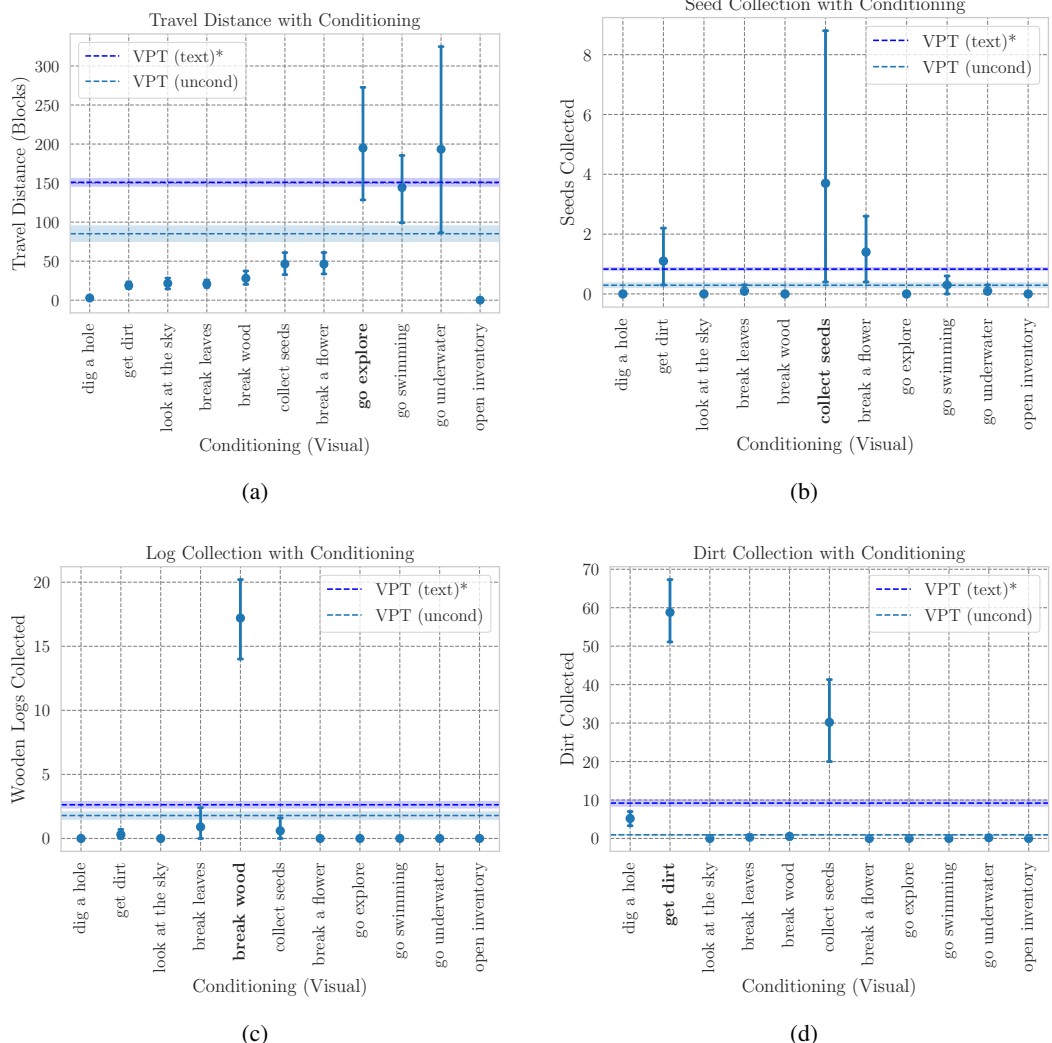

Figure 15: **Conditioning with Visual Goals**. We plot the performance of the programmatic metrics, along with their mean values and 95% confidence intervals, across different goal conditioning. See Figure 14 for visualization of these visual prompts. Plots are similar to Figure 20 in the VPT appendix [5]. Each conditioning variant is run with 10 trials, each with a different environmental seed and with an episode length of 3000 time steps (2.5 minutes gameplay). We use the policy that was trained using the hyperparameters specified in Table 1, and with conditional scaling values $\lambda = 7$. The dashed horizontal lines refer to an unconditional VPT agent ("*VPT (uncond)*") and a text-conditioned agent from the VPT appendix ("*VPT (text)\**") that was conditioned on the relevant text, for the purpose of comparison.

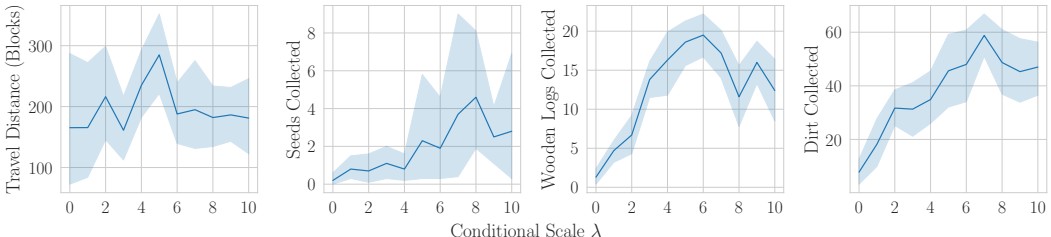

Figure 16: **Conditioning with Text goals**. See Table 3 for the exact text string used for each conditioning variant. We use the same policy model but with a conditional scaling value $\lambda = 6$. We observe strong steerability which outperforms text-conditioned VPT in Appendix I of [5], and we observe that prompt-engineering can improve performance.

Figure 17: The conditional scale $\lambda$ in classifier-free guidance [23] can be tuned to improve performance in each of the programmatic tasks. By tuning $\lambda$ to use classifier-free guidance at inference time, STEVE-1 is able to collect $7.5\times$ more dirt, $15\times$ more wood, $23\times$ more seeds, and travel $1.7\times$ further than at $\lambda = 0$ (no guidance).

| Task | Prompts | Precondition |
|---|---|---|
| dig a hole, get dirt, look at the sky, break leaves, get wood, get seeds, break a flower, go explore, go swimming, go underwater, open inventory | Use the best-performing prompts in Table 3 in the Appendix. | For grass and flowers, it works best when grass and flowers are in the current biome. We found breaking flowers to be less reliable than the others. |
| make a tower | "build a tower" | building blocks in hotbar. To obtain building blocks, you can use the dig a hole or get wood prompts from Table 3. |
| craft wooden planks | "make wooden planks, craft wooden planks" (*) | wooden logs. To obtain wooden logs, use the get wood prompt from Table 3. |
| place torches, place a crafting table, place wooden planks | "place [torches/a crafting table/wooden planks]" | [torches/crafting table/wooden planks] in the hotbar |
| make a crafting table | "make a crafting table" | wooden planks. |
| break stone | "mine stone, go mining, get stone" (*) | |
| get cobblestone | "mine stone, go mining, get cobblestone" (*) | pickaxe in hotbar. |
| create a wooden pickaxe | "craft a wooden pickaxe, make a wooden pickaxe" (*) | already looking at a placed crafting table, has necessary materials. |
| hit a sheep, hit a pig, hit a cow | "kill a [sheep/pig/cow]" | agent is close to and looking at the sheep/pig/cow. (See Appendix B.1). |

Table 4: Examples of tasks that STEVE-1 is able to achieve. (*) denotes that this is a single prompt-engineered prompt. Note that this list represents only a subset of the capabilities of STEVE-1 due to the difficulty associated with evaluating and describing the capabilities of open-ended models, which is an open research problem itself since capability depends strongly on preconditions, prompt engineering, and our own ability to come up with varied and challenging tasks. Please see Appendix B.4 for further discussion.

| Model | Simple Prompt | Complex Prompt |
|---|---|---|
| Stable Diffusion [50] | steampunk market interior | steampunk market interior, colorful, 3D scene, Greg Rutkowski, Zabrocki, Karlkka, Jayison Devadas, trending on ArtStation, 8K, ultra-wide-angle, zenith view, pincushion lens effect [20] |
| STEVE-1 | collect seeds | break tall grass, break grass, collect seeds, punch the ground, run around in circles getting seeds from bushes |

Table 5: Example of evolving simple prompts into more complex ones for various models.

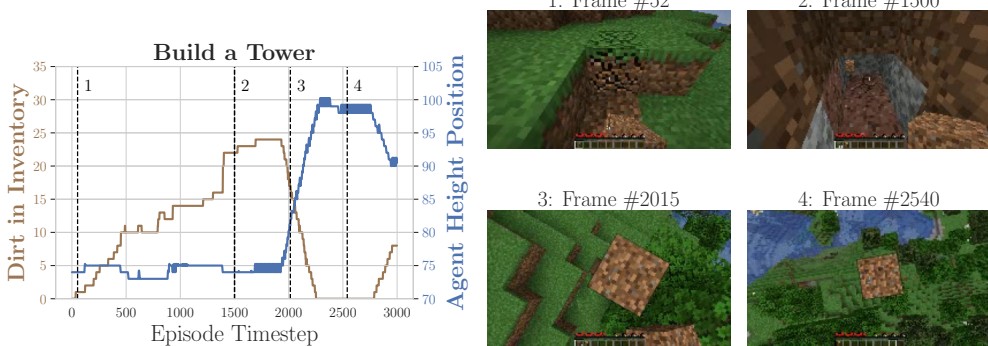

Figure 18: **Build a Tower task**. (*Left*) We track the amount of dirt in the inventory and the agent's height position (y-axis) throughout the episode. In the first 1500 timesteps, the agent is conditioned on the visual get dirt goal, then the agent is conditioned on the visual build a tower goal for the final 1500 timesteps. Vertical dotted lines with numbers indicate the corresponding frames on the right. (*Right*) The agent's observation frames at 4 different points in the episode. First the agent collects dirt, then begins to build the tower using the dirt blocks.

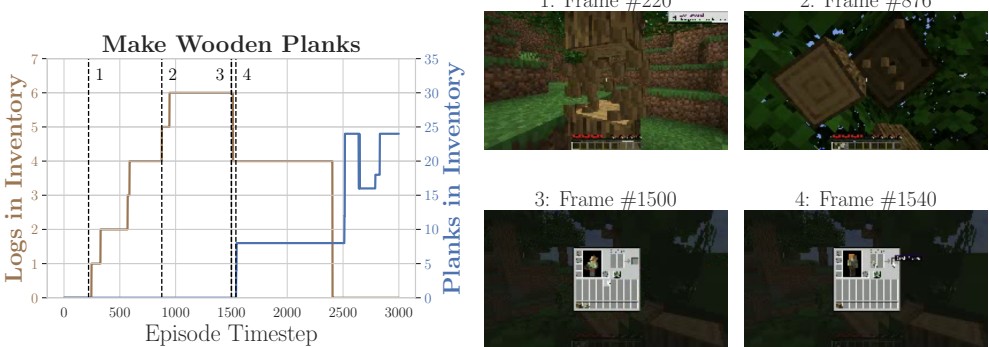

Figure 19: **Make Wooden Planks task**. (*Left*) We track the number of logs and planks in the inventory. In the first 1500 timesteps, the agent is conditioned on the visual break wood goal, then the agent is conditioned on crafting the visual wooden planks goal for the final 1500 timesteps. Similarly to Figure 18, a vertical dotted line annotated with a number indicates the corresponding frame to the right. (*Right*) The agent's observation frames at 4 different points in the episode. First the agent breaks trees to collect wooden logs, then opens the inventory and crafts wooden planks.

