# OpenReview forum: "STEVE-1: A Generative Model for Text-to-Behavior in Minecraft"
_NeurIPS.cc/2023/Conference — NeurIPS 2023 spotlight_

### Official Review · Reviewer_7QfK · 2023-06-27

**Soundness:** 3 good
**Presentation:** 3 good
**Contribution:** 2 fair
**Rating:** 5
**Confidence:** 5

**Summary:**

This paper proposes a generative pretraining method to learn an instruction-following agent in Minecraft by fine-tuning the VPT agent. It first trains an image-goal-conditioned policy and then leverages the foundation model MineCLIP as the bridge to map the language instruction into the image-goal space. Using this method, this paper claims that it can solve any short-horizon open-ended text and visual task in Minecraft.

**Strengths:**

(1) This paper explores the feasibility of jointly using the foundation models (such as the vision-language model MineCLIP and pre-trained policy model VPT) for solving decision-making problems and shows a preliminary conclusion.

(2) Using unCLIP approach to solve instruction-following decision-making problems is interesting.

(3) This paper proposes to improve the instruction sensitivity with the classifier-free guidance technique, which is interesting and reasonable.

(4) The paper is well-written and easy to follow.

**Weaknesses:**

(1) The paper overstates its performance by claiming that "STEVE-1 can follow nearly **any short-horizon** open-ended text and visual task in Minecraft."

To verify this, I downloaded the provided code and ran the attached checkpoints on some basic short-horizon tasks like "kill sheep", "kill cow", and "kill pig". Before testing these tasks, I had already summoned 10 "pigs", "cows", and "sheep" nearby. While I found that the agent actually killed some animals, it did not seem aware of which particular animal it was targeting. The behavior was more like "kill cow if there is a cow" rather than following the specific instruction. I tried several other prompt variants such as "hunt cow" and "hunt cow in Minecraft", but they were unsuccessful. Additionally, I tested STEVE-1 on some short-horizon crafting tasks with sufficient materials in the initial inventory, like "craft stick", "craft oak plank", and "craft torch". The agent merely opened its inventory and then acted randomly.

Given these observations, I do not believe that STEVE-1 has solved all short-horizon tasks, as it cannot even differentiate between "cows" and "sheep". This could lead others to misconstrue the extent of the research in the Minecraft environment.

(2) This paper lacks generalization experiments on unseen text instructions.

As the primary aim of this method is to support open-ended text instructions in STEVE-1, it is crucial to demonstrate its performance on unseen text instructions. The authors collected approximately 10,000 instruction-trajectory pairs, with 2,000 of them hand-labeled and an additional 8,000 instructions generated by GPT. It remains unclear whether the tasks ("dig dirt", "build a tower", "make wooden planks") in the experiments were already included in these instructions. If so, the experiments may not be particularly convincing, as neural networks can easily map the text instructions to corresponding visual embeddings in the training set. That is, pre-training STEVE-1 on visual instructions already achieves this goal and the later generative model training may be not important, which is not impressive. So, I strongly recommend including generalization experiments if the paper wants to show support for open-ended text instructions.

(3) The evaluation metric is insufficient.

The MineCLIP evaluation metric is not convincingly effective in measuring whether a trajectory corresponds to a given task. This is because the MineCLIP latent space cannot distinguish some trajectories well. For instance, we have found that it was unable to differentiate between two trajectories where one involved approaching a tree and the other involved moving away from it (MineCLIP was not sensitive to distance).

To address this issue, I recommend that the paper incorporates a success-rate-based confusion matrix, in which the values in Figure 3 (b) are replaced with the task success rate, conditioned on the given instruction.

**Questions:**

As stated in the weakness part.

**Limitations:**

The main limitations include (1) prompt design being too sensitive and (2) chain-of-thought prompting is not automated.
The authors have adequately stated the limitations.

---

> ### Author Rebuttal · Authors · 2023-08-10
>
> Thanks for the great questions and for engaging deeply with the work. We’re glad that you found the approach interesting and reasonable, and the paper easy to follow.  We address your questions and comments below.
>
> > The paper overstates its performance by claiming that "STEVE-1 can follow nearly **any short-horizon** open-ended text and visual task in Minecraft."
> >
>
> We agree that this overstates STEVE-1’s capabilities. Shortly after submitting we actually changed this language ourselves to say “follow a wide range of short-horizon text and visual instructions in Minecraft”. We hope you agree that this language more accurately describes the performance of STEVE-1.
>
> Regarding the specific examples listed (hitting a cow, crafting a stick), the behavior you noted seems about right for the weights that we uploaded to OpenReview. The phenomenon of the agent performing a seemingly related task rather than the intended one is something we noticed too and is related to the concept of goal misgeneralization (Langosco et al. 2022). Generalization can be helpful when the task we assign the agent is impossible to achieve from the current state and the agent instead performs a closely related action, but harmful when the task is achievable. We note two things: first, the powerful generalization ability of STEVE-1 probably comes from the MineCLIP embeddings and it especially improves the ability of STEVE-1 to follow visual instructions when the exact items or blocks nearby are not available in the current environment, which is an extremely common scenario. Second, we notice that the tendency of the agent to misgeneralize decreases with scale. For example, with a model trained on fewer data we find that asking the agent to look up and punch a tree to get a log often resulted in the agent looking in the air and punching nothing; training the model on more data results in the agent first walking over to a nearby tree and looking up to get a log.
>
> We are including (via the AC) an updated codebase with the ability to run the agent in an interactive mode where new instructions can be presented to the agent in real-time. This updated code also includes a new snapshot with updated hyperparameters (see general response) that shows a modest improvement in performance. We will add a section on goal misgeneralization and its relationship with scaling to the appendix of our updated paper, and we hope that future research can further investigate these issues and how to improve misgeneralization. We stress that for a wide range of tasks, including but not limited to the tasks included in our evaluations, STEVE-1 shows strong performance on both the original weights and the updated weights.
>
> > The [MineCLIP] evaluation metric is insufficient.
> >
>
> Please see the general response section titled “Reviewer 7QfK raised a concern regarding the strength of the MineCLIP evaluation.”
>
> > This paper lacks generalization experiments on unseen text instructions.
> >
>
> Thanks for pointing this out; we strongly agree that generalization experiments would improve the paper. During the rebuttal period we have performed a set of simple generalization experiments which we hope helps to answer these important questions.
>
> **Instruction Training Set Contamination:** Please refer to Table A in our rebuttal PDF. Among our evaluation instructions, the bolded instructions show up in the instruction-trajectory dataset. While some instructions do show up, most of the instructions do not show up in our training set (verbatim).
>
> **Training Set Decontamination:** To measure the effect on performance of removing a concept from the training set, we ran an experiment where we removed every instruction with the words “dirt” or “dig” in them and retrained the VAE model. This corresponds to around 10% of the instructions. We found that even without training on the concept of dirt or digging at all, STEVE-1 can still be instructed to dig holes and get dirt. This demonstrates clearly that STEVE-1 can generalize to unseen text instructions (see Figure B in rebuttal PDF) — likely because most of the text-understanding comes from the pretrained MineCLIP model which was trained on a highly diverse dataset of YouTube videos and captions. The prior VAE only needs to learn a simple mapping between the text and visual MineCLIP embeddings. Note that there is a slight decrease in performance across all tasks likely due to the smaller VAE training set (~10% less).
>
> The instruction-following capability of STEVE-1 is shared between: the policy, which learns to follow instructions in the visual MineCLIP embedding space; the MineCLIP text-encoder, which is trained to align well with the visual embeddings and performs most of the text-understanding; and our prior VAE model, which learns a simple function to translate between text and visual embeddings. While this means that the VAE doesn’t hold the most important role in giving instruction-following capabilities, we disagree with the reviewer that this is unimpressive. Contrary to this viewpoint, it is precisely our modeling setup which lets us fully exploit pretrained models such as MineCLIP to gain impressive language understanding without relying on having our own large datasets or compute.
>
> We conclude by stressing that while STEVE-1 is not perfect, it represents a large step towards a general recipe for creating generalist agents by building on pretrained models. As future generative visual-language/action models get more powerful and general, we think that approaches like STEVE-1, which can exploit and steer their capabilities, will provide great value and enable exciting future research.
>
> Thanks again for the thorough review. We hope that some of the additional experiments and explanations we have conducted in response to the issues that you raised go a long way towards improving the paper and changing your opinion. Please don’t hesitate to ask if you have any additional questions or require clarification.

---

> > ### Comment · Reviewer_7QfK · 2023-08-11
> >
> > Thanks for the author's rebuttal. I appreciate the author's honesty and additional experiments to solve my concern. I'm going to raise the final rating. I strongly recommend the final revision of this paper should include a more detailed limitation part (illustrate what steve-1 is not capable of and why).

---

### Official Review · Reviewer_vziy · 2023-07-07

**Soundness:** 3 good
**Presentation:** 3 good
**Contribution:** 3 good
**Rating:** 6
**Confidence:** 4

**Summary:**

This paper proposed an instruction-tuned model for Minecraft, which turns previous RL model like VPT into goal-conditioned model. The experiment shows that the proposed method can follow nearly any short-horizon open-ended text and visual task in Minecraft.

**Strengths:**

1. This paper proposes a novel technique for instruction-tuning video game RL-agent which is inspired by previous unCLIP method in the text-to-image generation field. This idea is interesting and the method can turn an RL agent into a goal-conditioned one which shows great potential for designing a more general RL agent.

2. The experiment part shows that the tuned agent can follow short-horizon goal instructions very well. When given the appropriate textual instruction, the proposed method collects 66x more dirt, 4.5x more wood, 28x more seeds, and travels 3x further than the unconditional agent.

**Weaknesses:**

This method is not applicable for long-horizon tasks in the open world. For example, in Minecraft, if the goal is obtaining a diamond, the instruction may be too difficult for the proposed method since it is not trained for long-horizon trajectories. Regarding this, it is not sure if this method can be applied to challenging tasks in the open world which may contain extremely long planning steps. In this paper, no complicated tasks like obtaining diamond are studied. Thus I am concerned the scalability of the proposed method to much complicated tasks.

**Questions:**

I am curious about how is the prompt designed in this paper? The detail of the prompt should be explained in the main text of the paper.

**Limitations:**

The limitation is adequately addressed in the paper. No discussion about negative social impact.

---

> ### Author Rebuttal · Authors · 2023-08-10
>
> Thank you for your review and comments. We’re glad to hear that you found our work interesting and that you see it as a potential method for designing more general sequential decision-making agents. Please see below for responses to your comments and questions.
>
> > This method is not applicable for long-horizon tasks in the open world. For example, in Minecraft, if the goal is obtaining a diamond, the instruction may be too difficult for the proposed method since it is not trained for long-horizon trajectories. … In this paper, no complicated tasks like obtaining diamond are studied. Thus I am concerned the scalability of the proposed method to much complicated tasks.
> >
>
> We agree that the STEVE-1 agent is currently not capable of accomplishing long-horizon tasks like obtaining a diamond. This is a limitation of our work and something that we think is an interesting direction for future work, as solving long-horizon tasks while taking actions using low-level mouse/keyboard controls is a very challenging and exciting research direction.
>
> Based on our experience, there are a few things that we think could help improve long-horizon performance:
>
> **1) Scaling:** Our scaling results indicate that as we train the agent on more data, more tasks become achievable. So scaling up the amount of data could enable the agent to complete longer-horizon tasks.
>
> **2) Finetuning with RL:** We also think that finetuning the STEVE-1 pre-trained agent with RL is another very interesting avenue. For example, VPT finetuned their pretrained agent with RL to learn to mine diamonds. So we think it’s likely that RL finetuning on top of the pretrained STEVE-1 could enhance long-horizon task completion abilities.
>
> **3) Using LLMs or VLMs:** Using LLMs or Visual-Language Models (VLMs) to automatically provide prompt chains to the STEVE-1 agent could be an effective way to improve long-horizon performance. We started to investigate this possibility with our prompt chaining experiments and we think that this is a very exciting direction for follow-up work.
>
> > I am curious about how is the prompt designed in this paper? The detail of the prompt should be explained in the main text of the paper.
> >
>
> Thanks for asking this question. We answer both how the prompts were designed and what the exact prompts are.
>
> - **Exact prompts:** Table 3 in the appendix includes a list of the full text prompts used for each task, but we agree that the exact prompts should be made more clear in the main text of the paper. We have included a modified version of the table that hopefully provides more clarity in exactly which text prompt corresponds to which evaluated task (see below). We will include this figure in future versions of the paper.
>
> | Figures 3 (Right) & 11 Label | Figures 13 & 14 Label | Text Prompt |
> | --- | --- | --- |
> | dig | dig as far as possible | dig as far as possible |
> | dirt | get dirt | get dirt |
> | sky | look at the sky | look at the sky |
> | leaves | break leaves | break leaves |
> | wood | chop a tree | chop a tree |
> | seeds | collect seeds | collect seeds |
> | flower | break a flower | break a flower |
> | explore | go explore | go explore |
> | swim | go swimming | go swimming |
> | underwater | go underwater | go underwater |
> | inventory | open inventory | open inventory |
> | dirt... | get dirt ... | get dirt, dig hole, dig dirt, gather a ton of dirt, collect dirt |
> | wood... | chop down the tree ... | chop down the tree, gather wood, pick up wood, chop it down, break tree |
> | seeds... | break tall grass ... | break tall grass, break grass, collect seeds, punch the ground, run around in circles getting seeds from bushes |
> - **Prompt design:** In our experiments we used both short and longer prompts. The short prompts are either taken from previous literature (e.g., the language-conditioning experiment in the VPT appendix) or they were simply the first thing we tried. The longer prompts were created by taking inspiration from the prompt engineering methods used with text-to-image models such as Stable Diffusion [47]. To design these prompts, we simply strung together a lot of terms related to our task in order to increase the specificity of the prompts. We were excited to discover that this style of prompt design inspired by the prompt engineering community works well in STEVE-1.
>
> Thank you again for the positive review. We are excited about the simplicity, scalability, and strong performance of STEVE-1 and the many future research opportunities that it unlocks. If you have any additional questions or needs for clarification, please don’t hesitate to ask.

---

> > ### Comment · Reviewer_vziy · 2023-08-21
> > **I will keep my rating.**
> >
> > Thanks to the authors' detailed feedback. After reading the rebuttal, I still have concerns about long-horizon ability but I do agree with some possible solutions proposed by the authors.

---

### Official Review · Reviewer_Q7hP · 2023-07-07

**Soundness:** 4 excellent
**Presentation:** 4 excellent
**Contribution:** 4 excellent
**Rating:** 7
**Confidence:** 3

**Summary:**

The paper introduces STEVE-1, a sequential decision-making agent designed to follow textual instructions and accomplish goals in the Minecraft environment. The authors utilize two pre-trained models, VPT (Video Pretraining Transformer) and MineCLIP, to facilitate this process. VPT is a transformer model trained to predict action sequences from aligned video sequences, while MineCLIP aligns consecutive video timesteps with corresponding transcripts in Minecraft.

To finetune VPT, the authors employ self-supervised behavioral cloning conditioned on latent visual goals. They generate goal-conditioned data by randomly selecting timesteps from episodes and using hindsight relabeling to set intermediate goals. MineCLIP is used to map textual goals to visual goal embeddings in the unCLIP-based approach. This mapping is achieved through a conditional Variational Autoencoder (VAE) with Gaussian prior and posterior, conditioned on MineCLIP text representations.

The training dataset comprises Minecraft gameplay data, combined with the OpenAI contractor dataset, and additional data generated using VPT. The authors curate 2,000 instruction-labeled trajectory segments, each consisting of 16 frames, and augment this dataset by identifying similar gameplay segments. Additionally, 8,000 additional instructions are generated using GPT-3.5 Turbo. Classifier-free guidance is employed to balance the logits between unconditional and conditional behavior.

The results demonstrate that STEVE-1 successfully solves various short-horizon open-ended text and visual tasks, with a training cost of only $60. The performance of the agent is evaluated using programmatic evaluation and MineCLIP evaluation, revealing significant improvements with goal conditioning, especially visual. The authors also observe that prompt chaining, as opposed to direct prompting, proves advantageous for complex tasks like building towers or creating planks. A thorough ablation study is conducted on the classifier-free guidance hyperparameter, pretrained VPT weights, and prompt engineering, providing valuable insights into the optimal settings for these components.

Overall, the paper showcases the effectiveness of STEVE-1 in achieving goals based on textual instructions in the Minecraft environment. The combination of VPT and MineCLIP, along with the conditioning techniques and ablation studies, contribute to a comprehensive understanding of the agent's performance and its potential applications.

**Strengths:**

- This unique adaptation - STEVE-1 - of the unCLIP method demonstrates the versatility and effectiveness of the approach in the context of sequential decision making in Minecraft.
- The experiments and analysis conducted in the paper are highly novel and insightful. The authors provide valuable findings, such as the benefits of prompt chaining, the potential for scaling to improve certain metrics, and the limitations observed in complex tasks. These insights enhance our understanding of the proposed approach and its implications.
- The paper demonstrates that prompt chaining is effective in accomplishing complex tasks, such as building towers or making wooden planks. The results show that success rates and programmatic evaluation metrics plateau after a certain number of frames, highlighting the potential and limitations of the approach. Additionally, the comparison to direct prompting reveals the superiority of prompt chaining in achieving satisfactory performance.
- The thorough ablation study conducted on various components, including the classifier-free guidance hyperparameter, pretrained VPT weights, and prompt engineering, provides valuable insights into the optimal settings for these elements.
- The inclusion of additional ablations, such as goal chunk sizes, VAE variants, and text augmentation, in the appendix further enhances reproducibility and facilitates a deeper understanding of the approach.

**Weaknesses:**

- The paper would benefit from clearer explanations regarding certain aspects, such as the distinction between packed hindsight relabeling and hindsight relabeling. Additionally, providing a more detailed explanation of the interpretation of the heatmap, specifically the meaning of the 1/0 values, would enhance understanding.
- Some questions (see section) during the review could be addressed in the main work or in an appendix to provide a more comprehensive understanding of the approach.
- The proposed approach combines existing ideas from well-established approaches, such as CVAE, MineCLIP, and VPT, which may limit the novelty of the work. However, the extensive experiments and detailed study conducted in the paper contribute to its significance and overall value.
- The scalability of the approach to other environments or datasets is not discussed or addressed in the paper. Considering the dependency on pretrained video-text aligned models and large-scale pretrained transformers like VPT, the applicability of the approach to environments without such models is unclear. Addressing this limitation and discussing potential scalability issues would enhance the practical relevance and broader applicability of the approach.

**Questions:**

- Can you provide insights into why random timesteps from episodes are selected as goals? Are there alternative approaches that could be considered, and why is this method deemed the most effective?
- Does randomly resetting the agent's memory and turning the agent to a random direction during data generation provide necessary benefits? What would be the implications if this step were omitted?
- How does the utilization of pre-trained VPT contribute to the approach when the input is modified with conditional goal embeddings? Can you elaborate on the specific advantages and improvements gained from incorporating VPT in this manner?
- Could you explain the process of generating additional text instructions based on GPT 3.5? How are these instructions generated and integrated into the training process?
- It is not clear why the graph from Baker et al. is not included in the text-conditioning results. Can you provide clarification or discuss the reasoning behind its exclusion?
- In Figure 4 (right), what is "relevant" and "irrelevant" in the graph? Clarifying the interpretation of these components would improve the understanding of the presented results.

**Limitations:**

- The authors extensively discuss the limitations of the approach, including challenges with multiple steps of reasoning, prompt engineering, and potential negative societal impacts. While further discussion on these aspects could be beneficial, there are no explicit omissions or significant limitations that require specific mention. The paper adequately addresses and acknowledges its limitations.

---

> ### Author Rebuttal · Authors · 2023-08-10
>
> Thanks for your constructive questions and feedback, and for recognizing the versatility, novelty, and significance of our work on STEVE-1. Please see below for responses to your comments and questions:
>
> > Clearer explanation on the distinction between packed hindsight relabeling and hindsight relabeling.
> >
>
> Hindsight relabelling was introduced in the Hindsight Experience Replay (HER) work [4]. With this technique, trajectories are relabelled with imagined goals, and these goals can be chosen using different strategies. *Packed hindsight relabeling* is our specific implementation of hindsight relabeling (see Algorithm 1 in Appendix), which “**packs” multiple relabeled goal sequences into a single sequence**. Specifically, we split a trajectory into multiple chunks and pick the last timestep in each chunk as the relabelled goal for that chunk (see Figure 2).
>
> > A more detailed explanation of the interpretation of the heatmap, specifically the meaning of the 1/0 values
> >
>
> Thanks for raising this. The ideal performance with the MineCLIP heatmap would be where the minimum values of each row lie on the diagonal. That is, the agent performs a specific task best when it is asked to perform that specific task. The numbers (1/0) on the diagonal correspond to the ranking that the diagonal has in its row. A 1 represents that the diagonal is the second lowest value in the row and 0 means it is the lowest.
>
> > The scalability of the approach to other environments or datasets is not discussed or addressed in the paper.
> >
>
> We designed STEVE-1 for Minecraft due to the availability of two key ingredients: 1) a strong behavioral prior (VPT), and 2) a powerful visual-language model which maps text and video to a joint embedding space (MineCLIP). However, the method used to create STEVE-1 is not specific to the Minecraft domain. Given the rapid development of generative models, we expect that similar models to VPT and MineCLIP will become available in many other domains. As these models become available, future work could investigate the applicability of the STEVE-1 approach to these other domains. Thanks for raising this key point, we will include a discussion of this in the final version of our paper.
>
> > Can you provide insights into why random timesteps from episodes are selected as goals?
> >
>
> We chose to randomly select future timesteps from episodes as goals primarily due to the simplicity of the approach, but also to ensure a diverse and unbiased coverage of potential goals achievable within the short horizon. By avoiding a specific heuristic strategy, we aim to prevent any potential biases in goal selection that might influence the agent's training, thereby promoting a more generalized performance. Future works can investigate the effects of alternative approaches that filter for semantically interesting timesteps as goals.
>
> > Does randomly resetting the agent's memory and turning the agent to a random direction during data generation provide necessary benefits?
> >
>
> Thanks for asking this question. These implementation details of the VPT dataset were chosen as heuristics to increase the diversity of the generated trajectories. Unfortunately, it’s very compute-intensive to get concrete answers about the effects that these decisions have. We will consider running additional experiments to investigate this question in the future.
>
> > How does the utilization of pre-trained VPT contribute to the approach when the input is modified with conditional goal embeddings? Can you elaborate on the specific advantages and improvements gained from incorporating VPT in this manner?
> >
>
> Great question. Yes, due to how we modify the inputs to the transformer, the input distribution is different from what VPT expects. However, since we finetune the modified VPT architecture on our gameplay dataset with relabelled goal embeddings, then the model learns to adapt to this new distribution. Empirically, in Figure 5 (left) we found that using the VPT pretrained weights improves performance as compared to finetuning from scratch. Note that this conditioning method via a bias to the transformer input was also explored in Baker et al.’s Appendix I. However, future works can study other mechanisms for conditioning.
>
> > Could you explain the process of generating additional text instructions based on GPT 3.5? How are these instructions generated and integrated into the training process?
> >
>
> We’ll update the appendix to include the GPT-3.5-turbo system prompt and user query prompt. Following the creation of approximately 8,000 additional examples from GPT, we further find potential matching clips from the gameplay dataset as described in Appendix D.2. Specifically, for each of our text labels, we find the top 5 closest (cosine similarity) timesteps in 2000 episodes from our gameplay dataset using MineCLIP’s similarity score. These 50,000 automatically-mined text-video pairs are added to the original 2,000 hand-labeled examples to form the final dataset used for training the prior. We will add an algorithm box to make this more precise in the final version.
>
> > It is not clear why the graph from Baker et al. is not included in the text-conditioning results.
> >
>
> We showed this comparison in Appendix E.2 (Figures 13 & 14) and we will add the results from Baker et al. to Figure 3 (left). The caveats with this comparison is that we have some differences in the experimental setup: (1) we only used half the episode length, and (2) it’s unclear exactly how Baker et al. defined their item count metric. Despite being evaluated for half the episode length, STEVE-1 achieves much stronger steerability performance.
>
> > In Figure 4 (right), what is "relevant" and "irrelevant" in the graph?
> >
>
> “Relevant” refers to the single relevant prompt for the task (”collect seeds” for task “Seeds Collected”, etc.) and “irrelevant” corresponds to the performance averaged over 10 other irrelevant prompts. We will clarify in the final version.

---

### Official Review · Reviewer_3aRk · 2023-07-08

**Soundness:** 4 excellent
**Presentation:** 4 excellent
**Contribution:** 4 excellent
**Rating:** 8
**Confidence:** 4

**Summary:**

Paper presents a method to create instruction following agents in Minecraft. It starts with collecting trajectories using OpenAI’s VPT Minecraft agents, which cannot be controlled through instructions. Then some intermediate visual observations are randomly selected as visual goals. These visual goals are then encoded by the visual encoder of a pretrained text-video contrastive foundation model in the Minecraft domain called MineCLIP. VPT is then fine-tuned to achieve these visual goals (specified as MineCLIP visual embeddings) using the same trajectories. Finally, a CVAE is trained to produce textual embedding that matches the visual goal embeddings, therefore the agents can be piloted by natural language instructions. Experiments highlight the performances of presented method on some entry-level tasks including dirt, log and seed collection. Additional results on prompt chaining, scaling, prompt engineering and other hyperparameters confirm the effectiveness of the proposed approach.

**Strengths:**

+The paper is overall clear and well-written. The presentation should be very friendly to readers without RL or LfD background by foundation model in general. The research problem here is relevant to the scope of NeurIPS and its proximity to many emerging topics including open-endedness, generalist agents and large models should be of interest to large audience of this conference.

+I find the approach presented here is technically sound with good results. Although the original VPT agent did offer some preliminary results on instruction following, but no enough details were provided so the method here seems to make solid contributions on building the first open-ended language-piloted agents in Minecraft. Some tricks like baseline subtraction seems helpful. Thanks for bringing this to the attention of the community.

+The authors did a good job highlighting go/no-go about their method. Just list a few that I find most interesting:

- MineCLIP evaluation, which help explain why their method works better on short-horizon task

- Prompt-chaining and metrics over time, which clearly demonstrate how the agent progresses under different prompt conditions.

- Prompt-engineering, which showcases how different instruction can have significant impact at the agent.


**Weaknesses:**

At this point, I don’t have any major concerns but here are a few suggestions:

-Baselines; in the main paper, I couldn’t find any baselines or ablations other than the main ”Steve-1” model and VPT(without instruction-following). I agree that there might not be many proper counterparts available but some single-task RL and imitation learning baseline, ex. [10] could still help with better sense on the actual gain the proposed model has in Minecraft. Moreover, the baseline subtraction trick should also be ablated in the main paper with more evidences on other tasks, it seems to be a working technique, but it must be better validated.

-Error bar; In Figure 6, I am not sure if the numbers in parentheses indicate error range. If they are not, error bar is needed for these results.

-Scaling: please elaborate more on “we see evidence of tasks that do not require much data for STEVE-1 to learn, tasks that steadily get more reliable as the agent is trained longer, and tasks where capability suddenly spikes after the agent reaches some threshold.” If not all tasks can benefit from scaling, why “Put together, this suggests that further scaling would likely significantly improve the agent”

-After reading the paper, it’s still unclear to me how the $60 budget is allocated. Please provide detailed explanation to justify this point.


**Questions:**

See [Weaknesses]

**Limitations:**

One thing that could be missing in the limitation statement is how well does the cVAE model perform on producing goal embedding and whether it could be a bottleneck to the overall performances on more challenging tasks like long-horizon planning, etc. The authors seem to blame MineCLIP for this but since in the original unCLIP paper, diffusion is used instead of a cVAE, more comparisons might be necessary.

---

> ### Author Rebuttal · Authors · 2023-08-10
>
> Thank you for your review and kind words. We are so glad that you found our work an approachable and valuable contribution to the community.
>
> > Baselines; in the main paper, I couldn’t find any baselines or ablations other than the main ”Steve-1” model and VPT(without instruction-following). I agree that there might not be many proper counterparts available but some single-task RL and imitation learning baseline, ex. [10] could still help with better sense on the actual gain the proposed model has in Minecraft.
> >
>
> The two most interesting baselines from our perspective are VPT [6] with text-conditioning (Appendix I in [6]) and the multi-task RL experiment done in the MineDojo paper [18]. As you mentioned, the VPT paper did not include many details and did not perform well. MineDojo evaluates on slightly different tasks and the authors appear to find very limited zero-shot generalization ability after training on 12 tasks. We’ll explicitly mention these two baselines in a new section of the paper and include a copy of their results. Note that Appendix E.2 does include a comparison to the VPT with text-conditioning results from Appendix I in [6], but we will include a more clear comparison in the main paper for future versions. Also, we have included a number of ablations in Section 4 and Appendix B that show the importance of our design choices, including experiments on the use of classifier-free guidance, pretraining, chunk size for packed hindsight relabeling, and VAE design choices. Thanks for pointing out [10]. We’ll explore it further.
>
> > Moreover, the baseline subtraction trick should also be ablated in the main paper with more evidences on other tasks, it seems to be a working technique, but it must be better validated.
> >
>
> We have updated the baseline subtraction trick figure (classifier-free guidance; Figure 5) to include our two other programmatic tasks of “travel distance” and “seeds collected”. We find that it does not make much of a difference for travel distance and improves the seed collection task. Please find the updated Figure C in our rebuttal PDF. Also, note that Appendix B.1 shows an ablation on the effect of using classifier-free guidance during finetuning training at all, which is different to testing the effect of the conditional scale parameter at inference time (Figure C in the rebuttal PDF).
>
> > Error bar; In Figure 6, I am not sure if the numbers in parentheses indicate error range. If they are not, error bar is needed for these results.
> >
>
> These are 95% confidence intervals. We have updated the figure caption to make note of this.
>
> > Scaling: please elaborate more on “we see evidence of tasks that do not require much data for STEVE-1 to learn, tasks that steadily get more reliable as the agent is trained longer, and tasks where capability suddenly spikes after the agent reaches some threshold.” If not all tasks can benefit from scaling, why “Put together, this suggests that further scaling would likely significantly improve the agent”
> >
>
> Thanks for the question. At a high level, we mean to say that since we see non-decreasing performance with scale (travel distance and seeds stay roughly constant, dirt and logs increase), then we posit that further scaling will help improve performance. We suspect that for tasks which didn’t benefit from scaling in our experiments, we are either close to optimal performance or we didn’t reach the critical amount of scale required to see strong performance. There is evidence of this type of sudden emergence in capability in both our dirt and logs task as well as in the literature [58]. We will improve the scaling section in the paper to clarify.
>
> > After reading the paper, it’s still unclear to me how the $60 budget is allocated. Please provide detailed explanation to justify this point.
> >
>
> The $60 cost we reported in the paper corresponds to the cost of renting a 8xA10g node using spot instances on AWS for 12 hours using our instances prices at the time. We are considering translating this into more standard numbers using on-demand pricing for A100s for the final paper release since we realized that spot prices can fluctuate and be misleading.
>
> > One thing that could be missing in the limitation statement is how well does the cVAE model perform on producing goal embedding and whether it could be a bottleneck to the overall performances on more challenging tasks like long-horizon planning
> >
>
> Good question. We don’t believe that the cVAE is a bottleneck to long-term planning since using STEVE-1 with visual instructions is equivalent to just bypassing the cVAE entirely and this doesn’t improve long-term planning abilities. However, a better cVAE model will likely improve text performance and we will add a discussion about this to the limitations section. Thanks for the suggestion.

---

> > ### Comment · Reviewer_3aRk · 2023-08-12
> >
> > Thank you for the reply. Some of my concerns have been addressed. But after reading other reviews as well, some additional questions pop up. Here are some follow-ups:
> >
> > -It would be much better off if you can pull some results from [10] and compare them with yours in the current submission. It seems to be a very recent and close counterpart (goal-conditioned control in Minecraft), and these numbers will help readers understand how open-world control can be improved with your techniques, especially with modern neural network architectures. Some additional discussions might be needed as well.
> >
> > -It's interesting to know you think of long-term tasks with steve-1, we may have to use visual instruction instead. Why is this the case? I suppose specifying a long-term task with text should be simpler, no? Also, what do you think is the bottleneck of steve-1 solving long-term tasks? I understand you've made it clear on the capacity of steve-1, which is mostly about short-term but I'm sure some analysis & prospect about solving long-term goals (this could be relevant to the prompt-chaining experiments in the main paper as well) in the limitation section should be of interest to reviewers and potential future readers.
> >
> > -I've read other reviews and I agree with Reviewer 7QfK that saying "STEVE-1 can follow nearly any short-horizon open-ended text and visual task in Minecraft" is improper given the current state of evaluation. I don't even think "a wide range of tasks" seems to be appropriate either. My suggestion is to make it clear what kind of tasks steve-1 is able to robustly accomplish by adding a table to the main paper, even better, putting it here in the comment section for the reviewer to examine as your code&model is also provided, and change your statement to something like "steve-1 is able to robustly complete <number of tasks> of tasks in Minecraft". Please avoid using vague and non-academic terms.

---

> > > ### Author Response · Authors · 2023-08-15
> > > **Response to Reviewer 3aRk [1/2]**
> > >
> > > > It would be much better off if you can pull some results from [10] and compare them with yours in the current submission. It seems to be a very recent and close counterpart (goal-conditioned control in Minecraft), and these numbers will help readers understand how open-world control can be improved with your techniques, especially with modern neural network architectures. Some additional discussions might be needed as well.
> > > >
> > >
> > > After reading through [10] carefully, we agree that it is an important related work and we thank you for bringing it to our attention. Both of our works focus on goal-conditioned control in Minecraft, with the major difference being that [10] trains on a fixed set of goals and STEVE-1 uses unCLIP, hindsight relabeling, and MineCLIP to learn goal-reaching behavior from a large dataset in a self-supervised way. We believe that many of the techniques in our work improve open-world control with modern neural network architectures, including our efficient packed hindsight relabeling implementation, classifier-free guidance, effectively using the knowledge in pretrained models with unCLIP, and our overall scalable recipe that learns to reach goals in a self-supervised way. We will be sure to include a discussion of this and the details of [10] (including its different goal-conditioning architecture) in the final version of the paper.
> > >
> > > Regarding comparing numbers, the results in [10] include the success rate for single-task training for “chopping trees” (50%), “combat cow” (58%), and “combat sheep” (60%). Since the task (both the world and the reward function), action space, observation space, and training data are very different, comparing to our own success rate numbers in Figure A of the rebuttal pdf is likely to mislead. Regardless, for most of the above mentioned techniques, we include ablation studies that show the improvements they can bring to open-world control.
> > >
> > > > It's interesting to know you think of long-term tasks with steve-1, we may have to use visual instruction instead. Why is this the case? I suppose specifying a long-term task with text should be simpler, no?
> > > >
> > >
> > > We are afraid there may have been a misunderstanding since we do not believe that there is a difference between visual and text instructions for long-term planning. We agree that specifying a long-term task with text should be simpler.
> > >
> > > > What do you think is the bottleneck of steve-1 solving long-term tasks? I understand you've made it clear on the capacity of steve-1, which is mostly about short-term but I'm sure some analysis & prospect about solving long-term goals (this could be relevant to the prompt-chaining experiments in the main paper as well) in the limitation section should be of interest to reviewers and potential future readers.
> > > >
> > >
> > > We think one bottleneck of solving long-term tasks is that during our packed hindsight relabeling, we limit the hindsight goals to at most 200 timesteps in the future (10 seconds). Due to this, tasks that require more than 200 timesteps to complete are technically out-of-distribution for STEVE-1. We experimented with using longer goal lengths (denoted ‘chunk size’) in Figure 10 in the appendix and found that the performance on all of our evaluation tasks tends to decrease if we increase this hyperparameter too much. We suspect that while an increased chunk size may be able to improve long-horizon performance, it also increases noise and comes at the cost of reducing performance on short-horizon goals. It is an important avenue for future work to investigate whether it is possible to achieve a better tradeoff.
> > >
> > > Improved performance on long-horizon tasks is one of the most important follow-up directions to improve upon STEVE-1. We point you to our response to reviewer vziy, where we enumerate a few other approaches that may improve long-horizon performance in future work:
> > >
> > > **1) Scaling:** Our scaling results indicate that as we train the agent on more data, more tasks become achievable. So scaling up the amount of data could enable the agent to complete longer-horizon tasks.
> > >
> > > **2) Finetuning with RL:** We also think that finetuning the STEVE-1 pretrained agent with RL is another very interesting avenue. For example, VPT finetuned their pretrained agent with RL to learn to mine diamonds. So we think it’s likely that RL finetuning on top of the pretrained STEVE-1 could enhance long-horizon task completion abilities.
> > >
> > > **3) Using LLMs or VLMs:** Using LLMs or Visual-Language Models (VLMs) to automatically provide prompt chains to the STEVE-1 agent could be an effective way to improve long-horizon performance. We started to investigate this possibility with our prompt chaining experiments and we think that this is a very exciting direction for follow-up work.

---

> > > > ### Author Response · Authors · 2023-08-15
> > > > **Response to Reviewer 3aRk [2/2]**
> > > >
> > > > > I've read other reviews and I agree with Reviewer 7QfK that saying "STEVE-1 can follow nearly any short-horizon open-ended text and visual task in Minecraft" is improper given the current state of evaluation. I don't even think "a wide range of tasks" seems to be appropriate either. My suggestion is to make it clear what kind of tasks steve-1 is able to robustly accomplish by adding a table to the main paper, even better, putting it here in the comment section for the reviewer to examine as your code&model is also provided, and change your statement to something like "steve-1 is able to robustly complete <number of tasks> of tasks in Minecraft". Please avoid using vague and non-academic terms.
> > > > >
> > > >
> > > > Thanks for raising this and we definitely agree that, especially in the context of providing a precise baseline to future works that compare with STEVE-1, avoiding vague statements is important. We will change the statement to “Our testing shows that STEVE-1 is able to robustly complete 12 of 13 goal-conditioned control tasks in our early-game evaluation suite.” (based on Figure A in the rebuttal pdf). It is important to recognize that these 13 tasks included in our evaluation are not strictly all the agent can do and that evaluating and describing the capabilities of open-ended models is an open research problem itself since capability depends strongly on preconditions, prompt engineering, and our own ability to come up with varied and challenging tasks (see the many recent works on the evaluation of LLMs for examples). We will add this discussion to our limitations section, change the language describing STEVE-1’s capability as described above, and include the following table where we summarize what, in our experience, STEVE-1 can do and how to reproduce it.
> > > >
> > > > Work-in-progress version of the table:
> > > >
> > > > | Task | Prompts | Notes |
> > > > | --- | --- | --- |
> > > > | Early-Game Evaluation Suite: dig a hole, get dirt, look at the sky, break leaves, get wood, get seeds, break a flower, go explore, go swimming, go underwater, open inventory | Use the best-performing prompts in Table 3 in the Appendix. | Precondition: For grass and flowers, it works best when grass and flowers are in the current biome. We found breaking flowers to be less reliable than the others. |
> > > > | make a tower | “build a tower” | Precondition: building blocks in hotbar. To obtain building blocks, you can use the dig a hole or get wood prompts from Table 3. |
> > > > | craft wooden planks | “make wooden planks, craft wooden planks” (*) | Precondition: wooden logs.  To obtain wooden logs, use the get wood prompt from Table 3. |
> > > > | place torches, place a crafting table, place wooden planks | “place [torches/a crafting table/wooden planks]” | Precondition: [torches/crafting table/wooden planks] in the hotbar |
> > > > | make a crafting table | “make a crafting table” | Precondition: wooden planks. |
> > > > | break stone | “mine stone, go mining, get stone” (*) |  |
> > > > | get cobblestone | “mine stone, go mining, get cobblestone” (*) | Precondition: pickaxe in hotbar. |
> > > > | create a wooden pickaxe | “craft a wooden pickaxe, make a wooden pickaxe” (*) | Precondition: already looking at a placed crafting table, has necessary materials. |
> > > > | hit a sheep, hit a pig, hit a cow | “kill a [sheep/pig/cow]” | Precondition: agent is close to and looking at the sheep/pig/cow. (See our response to Reviewer 7QfK.) |
> > > >
> > > > (*) note that this is a single prompt-engineered prompt

---

> > > > > ### Comment · Reviewer_3aRk · 2023-08-16
> > > > >
> > > > > Thank you for the very detailed reply. The comparison with [10], additional analysis on long-term tasks, and revised statement on the capacity of steve-1 all look good to me. Please ensure these contents will be included in the final version. I am happy to raise my score to 8.

---

### Author Rebuttal · Authors · 2023-08-10

Thanks to all the reviewers for your time and effort during the review process. We appreciate that you found our work well written, insightful, and novel, and we’re glad that there is excitement about our approach to creating an open-ended agent by building on pretrained models.

We have responded to each reviewer individually, uploaded a rebuttal PDF, updated our code (shared with the AC), and collected the below response to general concerns. If you find our answers responsive to your concerns, we would be grateful if you considered increasing your score, and if you have additional questions, we’re happy to engage further.



> Reviewer 7QfK raised a concern regarding the strength of the MineCLIP evaluation.
>

While we agree that MineCLIP is imperfect, our experimentation found that it performs impressively at understanding a wide variety of texts and relating them to Minecraft videos. Following the suggestion of reviewer 7Qfk, we have created a success-rate-based confusion matrix in order to demonstrate how MineCLIP evaluation aligns well with our human intuitions about task completion. This matrix is constructed using the exact same episodes used in the MineCLIP evaluation (Figure 3), but rather than use MineCLIP to evaluate task completion, we manually inspect each episode. We find that the overall pattern remains and STEVE-1 completes the instructed task the vast majority of the time (see the figure for the numbers). The success-rate-based matrix can be found in Figure A in the rebuttal PDF and we are considering replacing the MineCLIP evaluation matrix in the main paper with this new figure to increase clarity. Thanks for the suggestion!

> Reviewer 7QfK requested to see generalization experiments to unseen instructions.
>

We have included a new table showing the degree to which evaluation prompts appear in the training set for the VAE and we have also conducted a decontamination experiment where all instructions containing the words “dirt” or “dig” were removed from the training dataset. We find that the new model can still achieve tasks like digging and getting dirt even with this new setup, suggesting that the generalization capability of our agent to new instructions is strong and that most of the language-understanding capability comes from the pretrained MineCLIP model. We will include a discussion of these results in the updated version of the paper. Please see the response to reviewer 7QfK for more details.

> We have committed to making the following changes to enhance the paper in response to helpful comments by the reviewers. Many of these points are elaborated upon in responses to individual reviewers.
>
- Include in the main paper the VPT text-conditioning baseline and MineDojo multi-task RL baseline along with a discussion of limitations and differences.
- Include more tasks showing the effect of classifier-free guidance (baseline subtraction trick) in the appendix.
- Update the Figure 6 caption to indicate that the values in the parentheses are a 95% confidence interval.
- Improve the scaling section of the paper to make it more clear why we expect further scaling will improve agent performance.
- Add a section to the appendix indicating how we calculated the total cost of training STEVE-1.
- Add a discussion to the limitations section about the difference in performance between text and visual instructions.
- Add clarification regarding the differences between hindsight relabeling and packed hindsight relabeling.
- Add a more detailed explanation of the MineCLIP evaluation matrix.
- Add a section in the limitations section that summarizes what is needed to use our method in a new domain.
- Add a discussion of selecting goals in training randomly versus with a more sophisticated strategy.
- Add the prompts used to generate the GPT instructions for the prior training dataset.
- Update Figure 3 (left) to include VPT text-conditioning results as a baseline.
- Clarify “relevant” and “irrelevant” prompts in Figure 4 (right).
- Add a discussion on how the performance of STEVE-1 could be improved on longer horizon tasks.
- Add discussion about how the prompts were designed to the paper.
- Update Table 3 in the appendix to clarify which prompts correspond to which tasks in the MineCLIP evaluation matrix.
- Update “STEVE-1 can follow nearly any short-horizon open-ended text and visual task in Minecraft” to “STEVE-1 can follow a wide range of short-horizon text and visual instructions in Minecraft”.
- Add a discussion of goal misgeneralization to the main text.
- Add a human-labeled success-rate-based evaluation matrix to replace or in addition to the MineCLIP evaluation matrix in Figure 3 (right).
- Add discussion and experiments on training set contamination and generalization to new instructions to the appendix.

> Updated code and new model snapshots.
>

We have updated the code to fix a few bugs that we found and to facilitate running an interactive session with the agent. An interactive session makes it easy to test out and change prompts within an episode and to record videos. We also have included a slightly updated snapshot of STEVE-1 (new links in `download_weights.sh`). The main difference is that we trained the agent until the validation loss stopped going down (2 epochs → 3 epochs) and changed the VAE hyperparameters by increasing its size and tuning the $\beta$ hyperparameter. The performance is marginally better than the previous snapshot. This is the version of the agent and code that we will be releasing to the public as well as in the final NeurIPS supplemental materials. We’ve shared this anonymously with the AC.

We again thank the reviewers for their engagement and we appreciate all the suggestions that we believe will make the paper significantly stronger!

---

### Decision · Program_Chairs · 2023-09-21

**Decision:**

Accept (spotlight)

**Comment:**

The reviews are unanimous in favor of acceptance. We have also decided recommending it for spotlight. Considering that embodied agent is appealing to a broad community, we all agree that this paper is about some potentially very impactful topic. And one fundamental challenge in embodied agent is the text-to-action generation. The reviews all appreciate this paper’s conceptually simple formulation accompanied with thorough and careful analysis of prompt changing and potential of scaling. The reviews further appreciate the thorough ablation studies in the appendix and appreciate the careful documentation to facilitate reproducibility. However considering the diversity of complex tasks in Minecraft, we would also recommend this paper to include even more discussion about its limitation. Have we observed certain patterns when the proposed method certainly does not work? After careful discussion, one concern does remain. Considering this paper does require certain prior work at ready (e.g., VPT and mineclip), how a similar proposed method could work in other domains remains a question.